# Highly efficient light-emitting diodes via self-assembled InP quantum dots

Hui Li [1,2,7], Jingyuan Zhang[3,7], Wen Wen [2] ✉, Yuyan Zhao[2], Hanfei Gao[2], Bingqiang Ji [4] ✉, Yunjun Wang[5], Lei Jiang [1,2] & Yuchen Wu [1,6] ✉

Heavy-metal-free quantum dot light-emitting diodes (QLEDs) face commercialization challenges due to low efficiency and poor stability. Spin-coated quantum dot films often create charge leakage areas, limiting device performance. Here, we develop an evaporative-driven self-assembly strategy that enables the preparation of uniform and dense InP-based quantum dot films. During device operation, these films effectively suppress performance degradation caused by charge leakage. QLEDs with uniform and dense InP-based quantum dot films achieve high external quantum efficiency (26.6%) and luminance ($1.4 \times 10^5$ cd m$^{-2}$), along with considerable stability (extrapolated $T_{50}$ lifetime of 4026 hours at 1000 cd m$^{-2}$). For a $2 \times 3$ cm$^2$ InP-based device, the peak external quantum efficiency reaches 21.1%. By combining high-performance QLEDs with lithography technology, we fabricate miniaturized QLEDs with a minimum pixel size of 3 μm, achieving a resolution as high as 5080 pixels per inch and a peak external quantum efficiency of 22.6%.

Quantum dot light-emitting diodes (QLEDs) have emerged as strong contenders in solid-state lighting and display technologies due to their high color purity, efficiency, and wide color gamut[1–5]. The concept of QLEDs has been predominantly supported by II-VI group cadmium (Cd)-based quantum dots (QDs), due to their pronounced quantum confinement effects[6–8]. However, the toxicity of Cd-based QDs poses potential risks to the environment and human health, limiting their prospects for industrial applications[9]. Consequently, the development of heavy-metal-free QLEDs has become a focal point in the fields of display and lighting technologies[10–13]. Various techniques have significantly improved the external quantum efficiency (EQE) of heavy-metal-free QLEDs[9,13–16], including using hydrofluoric acid treatment to suppress trap states caused by surface oxygen on the core, developing core-shell structures to passivate traps, employing ligands to passivate surface defects, and optimizing charge transport layers for better energy level alignment. Currently, the EQEs of red, green, and blue heavy-metal-free QLEDs have reached 22.5%, 26.7%, and 24.7%,

respectively[13,15,17]. Nevertheless, compared to Cd-based QLEDs, heavy-metal-free red QLEDs still face challenges in terms of efficiency, stability, and luminance.

Although significant progress has been made in the synthesis of heavy-metal-free QDs, achieving uniform and dense QD films through spin-coating remains a challenge[18–20]. In the operation of heavy-metal-free QLEDs, significant charge leakage and low exciton radiative recombination efficiency are prevalent[21,22]. Therefore, developing effective strategies to fabricate dense and uniform QD films is crucial to address the issue of low efficiency in heavy-metal-free QLEDs. While multicomponent inks can suppress intense solutal Marangoni flows by lowering the solvent evaporation rate, challenges remain in achieving dense, uniform, and ordered nanoparticle films as the evaporation rate increases[23–25]. The addition of surfactants is an effective method for achieving uniform films, but it compromises device performance[26]. Langmuir-Blodgett technique can achieve both ordered assembly of QDs and the preparation of large-area films, but its compatibility with

[1]Key Laboratory of Bio-inspired Materials and Interfacial Science, Technical Institute of Physics and Chemistry, Chinese Academy of Sciences, 100190 Beijing, P. R. China. [2]State Key Laboratory of Bioinspired Interfacial Materials Science, Suzhou Institute for Advanced Research, University of Science and Technology of China, 215123 Suzhou, P. R. China. [3]Department of Physics, Shanxi Datong University, 037009 Datong, P.R. China. [4]School of Astronautics, Beihang University, 100191 Beijing, China. [5]Suzhou Xingshuo Nanotech Company, Ltd. (Mesolight), 215123 Suzhou, China. [6]University of Chinese Academy of Sciences (UCAS), 100049 Beijing, P. R. China. [7]These authors contributed equally: Hui Li, Jingyuan Zhang. ✉e-mail: wen24@ustc.edu.cn; bingqiangji@buaa.edu.cn; wuyuchen@iccas.ac.cn

device integration is limited[27]. Therefore, the fabrication strategy for QD films should meet the following three criteria: (i) suppressed uncontrollable fluid flows to form large-area, uniform, and dense films, (ii) stabilized the motion of the three-phase contact line (TPCL) to achieve long-range ordered arrangement of QDs, and (iii) compatibility with device integration.

Here, we develop an evaporation-driven self-assembly strategy for manufacturing large-area, uniform, dense red InP-based QD films. This self-assembly strategy consists of three-ended closed micropillar templates for anchoring the liquid, manipulating the TPCL of the QD solution to achieve directionally dewetting in a straight state, and then assembling the QDs into a large-area, uniformly dense red InP-based QD film. Using this integrated approach combining QD liquid-phase assembly and QLED fabrication, we observe high electroluminance (EL) performance in QLEDs, including: (i) significant inhibition of charge leakage during QLED operation, (ii) high EQE (peak EQE = 26.6%) and luminance (peak luminance = $1.4 \times 10^5$ cd m$^{-2}$), (iii) operational lifetime ($T_{50}$ @ 1000 cd m$^{-2}$ = 4026 h) close to commercial, (iv) peak EQE of 21.1% for $2 \times 3$ cm$^2$ InP-based QLEDs, and (v) integration with micro-QLEDs, achieving a minimum pixel size down to 3 μm (corresponding to 5080 pixels per inch (PPI)) and a peak EQE of 22.6%. The further development of heavy-metal-free QD self-assembly technology enhances the efficiency, stability, and scalability of QLEDs, providing a pathway for the commercialization of environmentally friendly display and lighting devices.

## Results

### Uniform and dense quantum dot films suppress charge leakage

In addition to dense hole transport layers (HTLs)[2], fabricating uniform QD films serves as another crucial strategy to suppress charge leakage, loss, and thermal generation during the operation of QLEDs. Rough QD films introduce numerous charge leakage regions and additional non-radiative recombination sites, leading to energy dissipation in the form of heat[3,28]. Moreover, leaked electrons reach the HTL, causing energy release in the form of heat at the interface between the HTL and QD layer, resulting in parasitic emission in the HTL[3], as shown in Fig. 1a. In contrast, uniform QD films prevent direct contact between the HTL and electron transport layer (ETL), thereby impeding electron transition to the HTL, suppressing heat generation at the interface between the HTL and QD layer and parasitic emission in the HTL. Furthermore, uniform QD films facilitate a more even and rational distribution of charge carriers, reducing the probability of carrier loss and non-radiative recombination, as shown in Fig. 1b.

In this study, we introduce an evaporation-driven QD self-assembly strategy, as shown in Fig. 1c. During solvent evaporation, the liquid is anchored by the micropillar template, guiding the TPCL into a straight line and directing its retraction from the open end[29,30]. As the TPCL slides, QDs assemble spontaneously at the TPCL on the substrate side, resulting in a uniform and dense QD film[31,32]. Compared to spin-coating deposition, our approach provides a slower and more controlled solvent evaporation rate, while disordered internal liquid flows are suppressed.

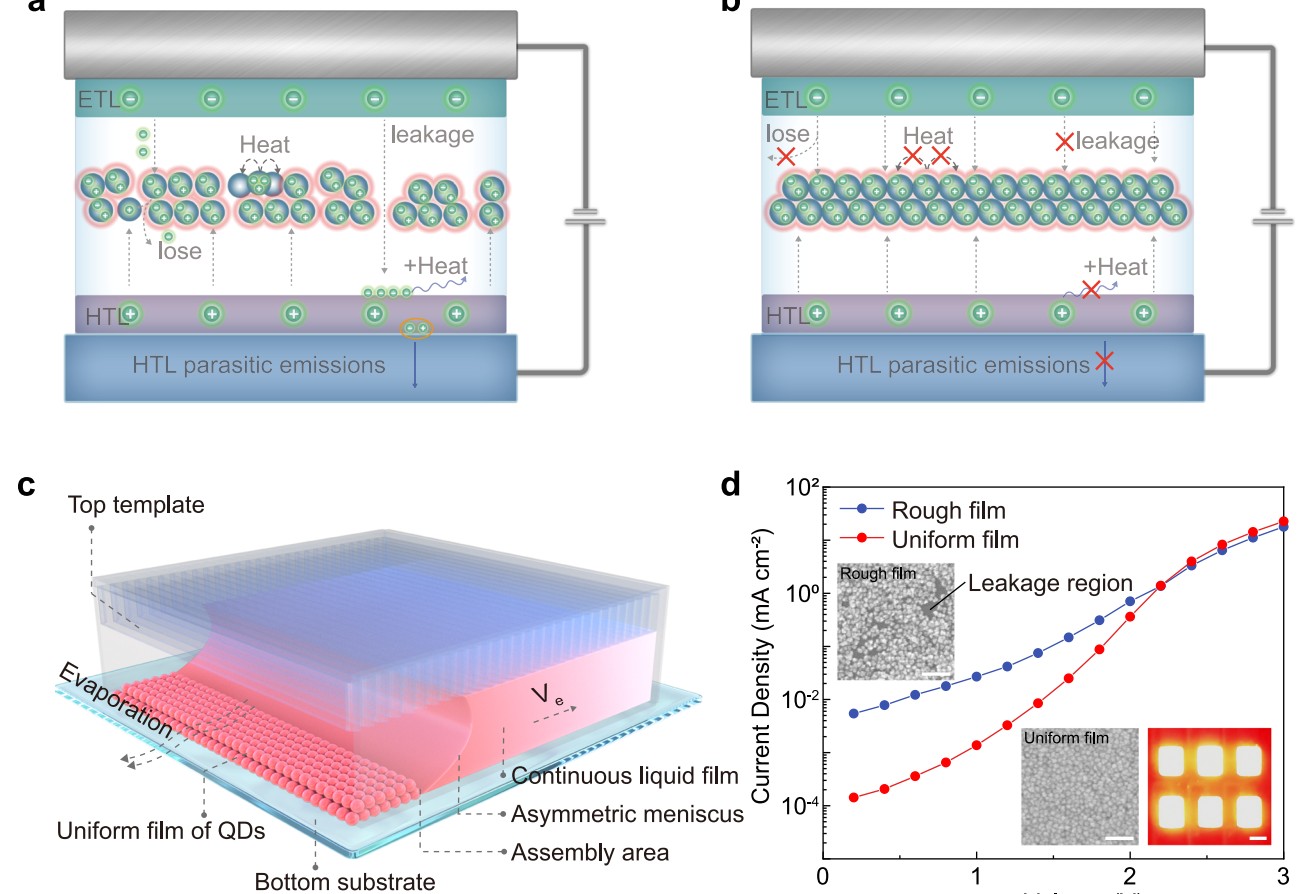

**Fig. 1 | Influence of the uniformity of QD films on the operation of QLED.**
**a** Schematic diagram illustrating the influence of rough QD films on carrier transport and recombination in QLED operation. **b** Schematic diagram illustrating the influence of uniform QD films on carrier transport and recombination in QLED operation. **c** Schematic diagram depicting the self-assembly of QDs driven by evaporation of liquid meniscus. **d** Current density-voltage curves of two QLEDs in the ohmic current region. Insets: SEM images of rough and uniform QD films. Scale bar, 100 nm. Microscopic image of the QLED in operation. Scale bar, 1 mm.

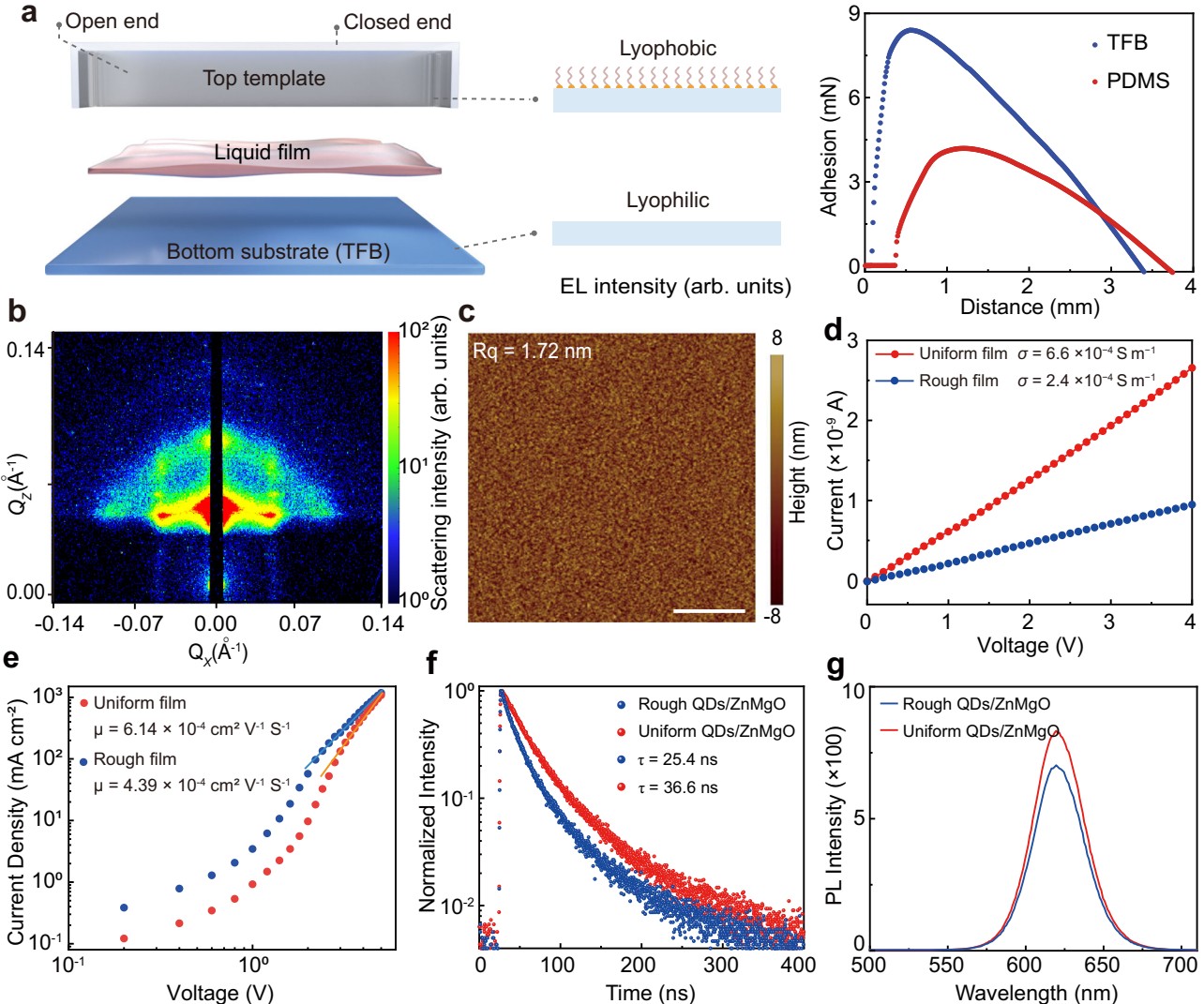

**Fig. 2 | Fabrication and characterization of uniform QD films. a** Anti-adhesive modification of the micropillar template. **b** GISAXS graph of InP-based QD self-assembled films. **c** Surface flatness map of InP-based QD self-assembled films. Scale bar, 1 μm. **d** Conductivity comparison based on I-V characteristics of two QD films. **e** Electron mobilities measured by space-charge limited current. **f**, **g** Transient fluorescence decay dynamics and steady-state fluorescence spectra of two types of QD films on ZnMgO/QD/substrate structures.

To investigate the influence of QD deposition morphology on charge transport in QLEDs, we adjusted the deposition strategy of QDs and prepared two types of QD films: one based on spin-coating deposition methods (rough, Fig. 1d inset, top left), and the other based on a self-assembly strategy (uniform, Fig. 1d inset, bottom right). After integrating these two types of QD films into QLEDs, we observe that the current density-voltage curves in the ohmic current region exhibited an approximately linear relationship (Fig. 1d). The current density of QLEDs based on uniform QD films is an order of magnitude lower than that of QLEDs based on rough QD films.

## Self-assembly and characterization of quantum dot films

To ensure an ordered deposition of QDs on the bottom substrate, we perform anti-adhesive modification of the micropillar template using dimethyldichlorosilane (DMS)[33]. A video clearly demonstrates that QD droplets slide off the anti-adhesive interface without observable residues (Supplementary Movie 1). The static contact angle (CA) of octane on the modified surface of the micropillar template is 24° with an adhesion force of 4.19 mN, while TFB surface exhibits a smaller static CA of 6.7° with a larger adhesion force of 8.4 mN (Fig. 2a and Supplementary Fig. 2). To identify the role of micropillar templates in high-

quality QD film preparation, we simulate the dewetting process of the solution (Supplementary Fig. 3). The difference in contact angle at the bottom substrate and top template creates an asymmetric meniscus, where the TPCL at the substrate lags behind that at the template.

Under the influence of the surface tension force at the TPCL, the continuous liquid film exhibits a pinning effect at the straight edge of the micropillars[34]. Since the liquid film is confined in a narrow gap with a thickness much smaller than the width, the retraction of liquid film becomes a 2D problem, and thus the TPCL on the TFB substrate also follows a straight line parallel to the micropillar edge. While on the substrate paired with a template without micropillars as well as the pinning effect, the TPCL exhibits a curved line due to the wall effects at two sides (Supplementary Fig. 4a). To mitigate the spreading of octane along the side walls, we design a reverse-curved structure at both ends of the micropillar template, finally yielding a neat straight assemble edge (Supplementary Fig. 4b)[35]. As the solvents evaporate and the liquid film dewets, the TPCL jumps from edge to edge at the micropillars and retracts in a stick-slip way, as observed during droplet evaporation process at pillared surface[36,37]. This stick-slip behavior results in the fluctuations of the TPCL on the bottom substrate. To better evaluate the impacts of these fluctuations on QD self-assembly,

we introduce the concept of error rate, defined as the ratio of fluctuation time to the total dewetting time within one stick-slip cycle. Based on regions with an error rate below 5% (Supplementary Fig. 4c), we design a micropillar template with a width of 20 μm, a spacing of 1 μm, and a depth of 3 μm for QD self-assembly (Supplementary Fig. 5), which facilitates a straight stably advancing assemble edge (Supplementary Fig. 6).

Equally important, our design also makes sure the deposition and assembly of QDs only at the bottom substrate, while maintaining a clean top template as the liquid film dewets. The strategy lies in the stable internal directional flow resulted by the uneven evaporation rate along the asymmetric meniscus, created by the wettability difference between top and bottom plates. It is well known that the evaporation rate is largely enhanced at the wedge of geometry near the TPCL, which increases significantly with decreasing contact angle, indicating a larger evaporation rate near the TPCL at the bottom substrate[38–40]. Additionally, the meniscus near the top template is far from the ambient and thus need to overcome a higher vapor concentration in comparison, further lowering down its evaporation rate. As the evaporation flux across the meniscus divergences at the bottom substrate, an internal capillary flow is triggered towards the bottom meniscus to compensate the liquid loss[41]. More importantly, the uneven evaporation along the meniscus creates a temperature gradient from bottom to top surfaces, yielding an opposite surface-tension gradient for triggering a downward thermo-Marangoni flow along the meniscus[42]. Under the influences of the evaporation-driven capillary flow and the thermo-Marangoni flow, the particles well dispersed in the solvent enrich at the wedge region near the bottom TPCL, depositing and assembling as the TPCL retracts (Supplementary Fig. 7). This phenomenon is similar to the coffee ring effect in an evaporating sessile droplet with downward interfacial Marangoni flow[43–45]. However, different from the chaotic internal flows with uncontrollable and non-regular TPCL (assembly edge) in evaporative droplets, our design facilitates a directional and stable 2D internal flow and assembly edge, which ensures regulated mass transfer along the width direction (Supplementary Fig. 4a)[46].

The particles near the TPCL can assemble in order if they have enough time to organize themselves within the strain by Brownian motion. The comparison between the Brownian time scale ($r_p^2/D_p$) and the flow time scale ($L/U$) can be represented by the Peclet number $Pe = Ur_p^2/(LD_p)$[25,40], where $r_p$ is the particle radius, $L$ is the initial interparticle spacing, $D_p = K_BT/(6\pi\mu r_p)$ is the particle diffusivity calculated by the Stokes-Einstein relation, and $U$ is the characteristic flow velocity represented by the TPCL retraction velocity ($< O(10^{-7})$ m s$^{-1}$). Using the parameters in our experiments, we obtain $Pe < O(10^{-6}) \ll 1$, which means that the QDs indeed have enough time to form an ordered film. At room temperature of 25 °C or lower, the solvent evaporation as well as TPCL retraction are mild and the QDs assemble is highly ordered, and we can observe that the roughness of the prepared QD films is lower compared to that at elevated temperatures. The thickness of the film can be adjusted by varying the concentration of the QD solution (Supplementary Fig. 8).

We select red InP-based QDs for film fabrication (Supplementary Fig. 9) because they represent the most promising alternative to Cd-based red QDs. During the dewetting process, Supplementary Figs. 6, 10a illustrates the straight geometry of the TPCL, ultimately achieving the preparation of a 25 cm² film (Supplementary Fig. 10b). Scanning electron microscopy (SEM) images reveal that the QD films fabricated through the self-assembly strategy are denser and more uniform (Supplementary Fig. 11a). The small-angle X-ray scattering (GISAXS) images of the self-assembled QD films show clear diffraction spots, indicating that the assembly of the QDs is highly ordered (Fig. 2b and Supplementary Fig. 12b). Atomic force microscopy (AFM) images demonstrate that the roughness of the self-assembled QD films is low, measuring only 1.72 nm (Fig. 2c and Supplementary

Fig. 12a). The uniformity of the electrical conductivity further suggests a more rational distribution of charge carriers (Supplementary Fig. 11b). Confocal imaging microscopy and hyperspectral Raman imaging further confirm that the self-assembled QD films exhibit uniform fluorescence emission intensity across a range of 100 μm to 1 cm (Supplementary Figs. 11c–e). These results indicate that the QD films produced using the self-assembly strategy possess higher density and uniformity. The consistency in uniformity and density of QD films made with n-hexane, cyclohexane, and toluene corroborates the broad applicability of the self-assembly strategy across various solvents (Supplementary Fig. 14). The enhanced conductivity (Fig. 2d), electron (Fig. 2e) and hole (Supplementary Fig. 13) mobilities measured by space-charge limited current, and interface potential (Supplementary Fig. 11f) indicate that self-assembled QD films exhibit more efficient carrier transport than spin-coated films, validating densely packed QDs. Figure 2f, g shows the time-resolved PL and steady-state PL spectra of self-assembled and spin-coated QDs interfaced with ZnMgO. Higher PL intensity, in line with longer PL lifetime, observed in self-assembled QDs indicates that the densely packed self-assembled QD films can effectively prevent ZnMgO penetration and PL quenching. These characteristics facilitate high-performance red InP-based QLEDs.

## Device performance characterizations

Next, we demonstrate substantial improvement in the EL performance of InP-based QLEDs by employing a uniform QD film as the emitting layer. We integrate two types of QD films into QLED devices (ITO/PEDOT:PSS/TFB/QDs/ZnMgO/Al), as shown in Fig. 3a. Cross-sectional transmission electron microscopy (TEM) images of the devices are shown in Fig. 3b and Supplementary Fig. 15, revealing the thickness of each functional layer through elemental distribution. Utilizing ultraviolet photoelectron spectroscopy (UPS), we determine the valence band maximum (VBM) and conduction band minimum (CBM) positions of the QD films (Fig. 3c and Supplementary Fig. 16). Figure 3d presents a comparison of the EL spectra of QLEDs based on the two types of QD films at the same current density (~600 mA cm⁻²). As the uniformity and density of the QD film increase, the parasitic emission intensity from the HTL gradually decreases and eventually disappears (Supplementary Fig. 17)[2,3].

Based on the current density-luminance-voltage characteristics of QLEDs with two different types of films (Fig. 3e), the QLED with a uniform film exhibits a lower injection current before turn-on, indicating that the dense and uniform QD film can suppress leakage current. Although the turn-on voltages of the two devices are similar (2.0 V), at the same current density (~2.4 V, ~3.6 mA cm⁻²), the luminance of the QLED with the uniform film is about twice that of the QLED with the rough film. Due to the increased luminance, the QLED with the uniform QD films have achieved unprecedented EQE of 26.6% and luminance of $1.4 \times 10^5$ cd m⁻² (Supplementary Table 2). In comparison, the QLED with the rough QD films had a peak EQE of 18.9% and a maximum luminance of $6.1 \times 10^4$ cd m⁻² (Fig. 3f). The histogram of Fig. 3g shows the peak EQE statistics (30 samples) for both two types of QD films, with an average EQE of 22.3% for QLEDs with the uniform QD films. To further understand the mechanism of performance enhancement, we measure the capacitance-voltage characteristics of QLED devices (Supplementary Fig. 18a). The QLEDs with the uniform, dense QD films exhibit lower capacitance across the entire voltage range, indicating suppressed charge accumulation. Additionally, we conduct transient electroluminescence (TrEL) measurements on QLEDs (Supplementary Fig. 18b). The comparison of TrEL kinetics reveals that QLEDs with uniform, dense QD films exhibit shorter turn-on times and faster TrEL. These results indicate that QLEDs with uniform, dense QD films not only achieve higher electron-hole radiative recombination rates but also significantly suppress charge leakage.

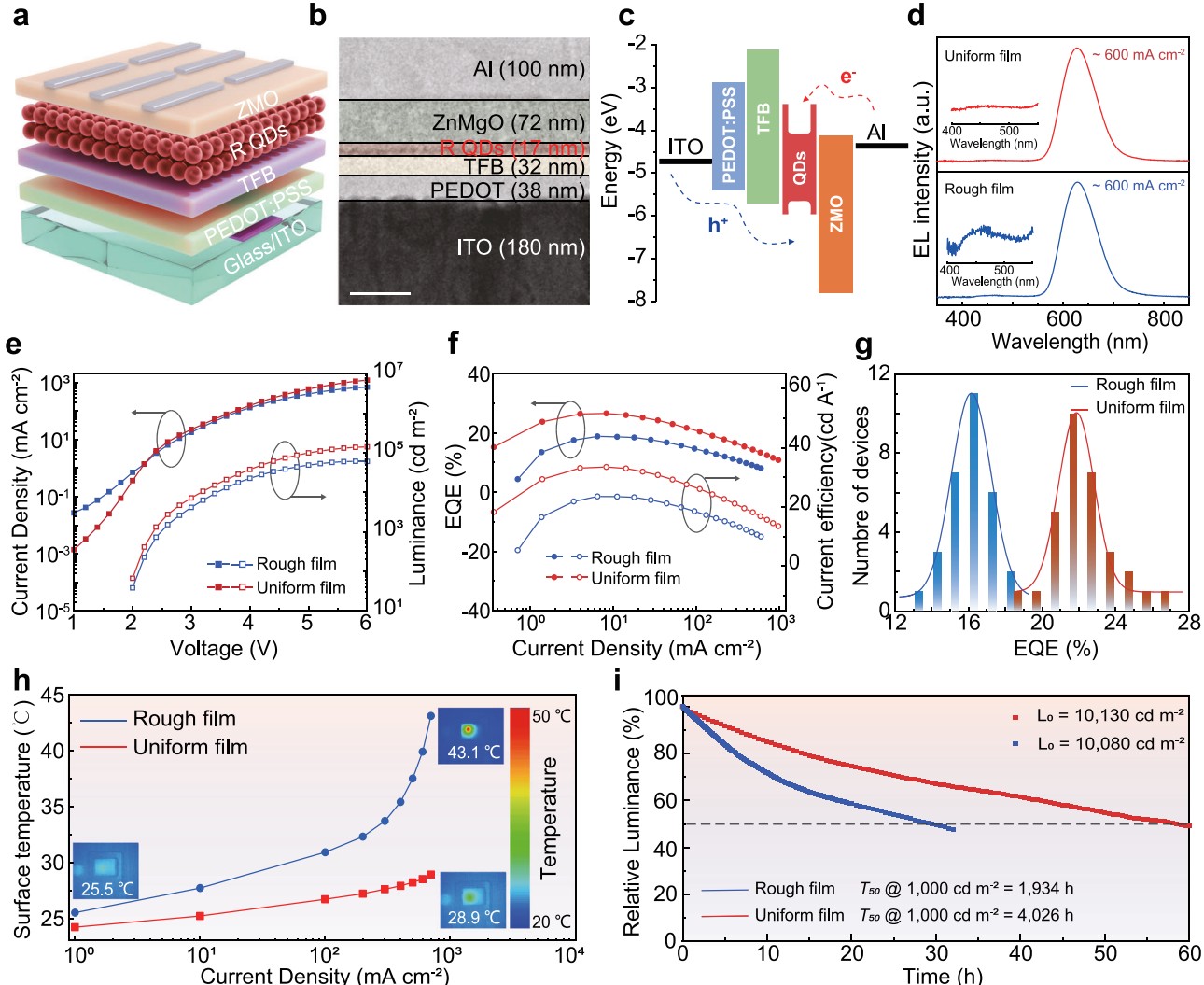

**Fig. 3 | Presents experimental studies of InP-based QLED devices. a** Schematic diagram of the QLED device structure. **b** TEM image showing a cross-section of the QLED device structure. Scale bar, 100 nm. **c** Energy level diagram of the QLED and the various functional layers of the device. **d** EL spectra of QLEDs based on rough and uniform QD films at the same current density (~600 mA cm⁻²). The spectral intensity of EL is normalized and logarithmically treated. Insets: zoomed-in views of EL spectra from the HTL. **e, f** current density-luminance-voltage and EQE-current efficiency-current density characteristics for the two QLEDs. **g** Histograms showing the Peak EQEs of the two types of devices. **h** Evolution of surface temperature under different current densities for the two QLEDs. Insets shows infrared images of the two devices at current densities of 1 mA cm⁻² and 700 mA cm⁻². **i** Relative luminance-time characteristics of the two QLEDs.

The uniform QD film plays a crucial role in suppressing heat generation and extending the operational lifespan of QLED devices. We utilize infrared thermal imaging to monitor the temperature variations of QLEDs during operation. Prior to activation, both two types of QLEDs exhibit surface temperatures of 23.1 °C. As the current density increases, we observe that the surface temperature of QLEDs with the rough QD film increases faster than that of the uniform films (Fig. 3h). At the current density of 700 mA cm⁻², the surface temperature of QLEDs with the rough QD film is 14.2 °C higher compared to those of uniform films. Moreover, QLED devices with the uniform QD film demonstrate prolonged operational lifespans. At an initial luminance of 10,080 cd m⁻², the $T_{50}$ lifetime of QLEDs with the rough QD film is 29.52 h, while for QLED with the uniform films at an initial luminance of 10,130 cd m⁻², the $T_{50}$ lifetime is 58.52 h (Fig. 3i). Extrapolating the $T_{50}$ lifetime of these devices at an initial luminance of 1000 cd m⁻² using the relation of $L_0^n T_{50} = \text{const}$ yields approximate values of 1934 h and 4026 h for spin-coated and self-assembled devices, respectively. Here, $L_0$ represents the initial luminance, $T_{50}$ denotes the corresponding half-life, and $n$ signifies the acceleration factor, with values of 1.81 and 1.82, respectively (Supplementary Fig. 19).

## Universality of the self-assembly strategy

To confirm the versatility of the evaporation-driven self-assembly strategy, we fabricate QLEDs by self-assembly of InP-, ZnSeTe- and Cd-based QDs. All of these devices show significant enhancement of performances compared to the spin-coated films (Supplementary Figs. 20, 21). To validate the scalability of the self-assembly strategy, we fabricate a red InP-based QLED with an effective area of 2 × 3 cm² and a high-resolution red InP-based QLED (Fig. 4a, b). The large-area red InP-based QLED exhibits a uniform electroluminescent surface, and the current density-luminance-voltage curves demonstrate that the large-area QLED maintain a low turn-on voltage characteristic. Additionally, this device achieves a peak EQE of 21.1%, retaining 80% of the efficiency compared to devices in an area of 0.03 cm². In the high-resolution red InP-based QLED, the single-pixel size is reduced to 3 μm, achieving a resolution of 5080 PPI. This device presents a peak EQE of 22.6%, maintaining low leakage current characteristics. Furthermore, the luminance of these devices retains 90% of that of 0.03 cm² devices. To demonstrate the practical applications of the self-assembly strategy in patterned QLEDs, we employ this method to deposit QD thin films onto a photoresist blocking layer (Fig. 4c). We defined

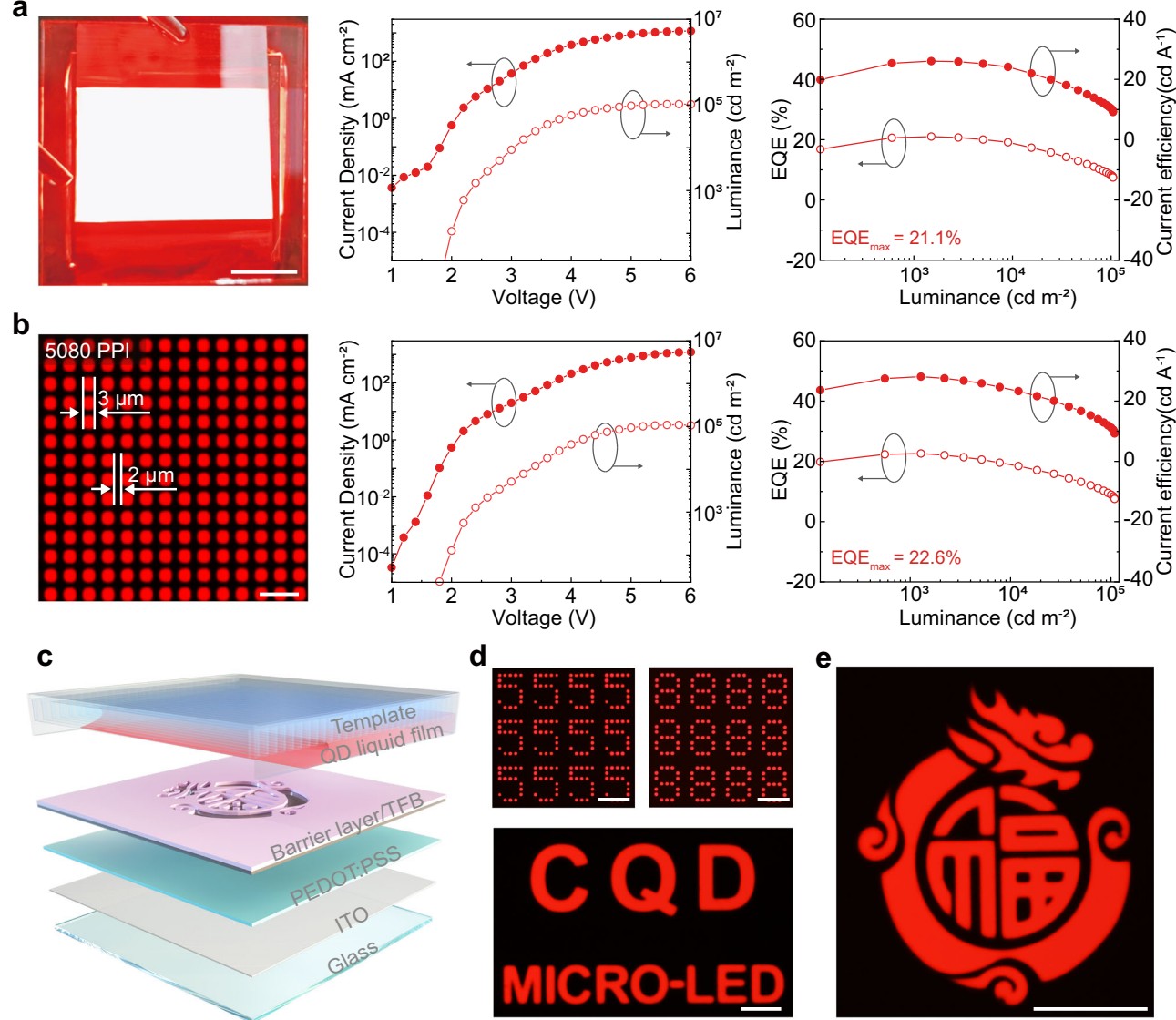

**Fig. 4 | Large-area and Patterned QLEDs. a** Photograph of a large area ($2 \times 3\ cm^2$) red InP-based QLED and its current density-luminance-voltage characteristics and EQE-Current efficiency-luminance characteristics. **b** Pixel photos of high resolution red InP-based QLED (Top right: 5080 PPI resolution) and its current density-luminance-voltage characteristics and EQE-current efficiency-luminance characteristics. **c** Schematic illustration of patterned QLED device fabrication. **d**, **e** EL static image of InP-based QLEDs. Scale bar, 1 cm (**a**), 10 μm (**b**), 100 μm (**d**), 500 μm (**e**).

miniaturized LED pixels by creating microholes onto the photoresist layer through a photolithography process. The thickness of photoresist layer is controlled to 20 nm to enable the continuous sliding of TPCL during QD self-assembly (Supplementary Fig. 22, 23). Consequently, this approach successfully creates passive-matrix QLEDs for displaying static images (Fig. 4d, e). These results highlight the potential of the self-assembly strategy in the production of high-performance red InP-based QLEDs and other QD-based optoelectronic devices.

## Discussion

In summary, this study underscores the pivotal role of uniform QD films in enhancing the EL performance and operational stability of QLEDs. By employing an evaporation-driven QD self-assembly strategy, we achieve large-area, uniformly dense QD films. Comparative analyses between self-assembled and spin-coated QLEDs demonstrate significant improvements of current density, luminance, and EQE in densely packed self-assembled QDs. Furthermore, infrared thermal imaging demonstrates reduced heat generation and prolonged

operational lifetime in QLEDs with the uniform QD film. The universality of the self-assembly strategy is validated through experiments on Cd-based QLEDs, showcasing promising results in terms of peak EQE and color gamut coverage. Overall, this work highlights the importance of microfluid-assisted QD self-assembly techniques in advancing the performance and applicability of QLED technology, particularly in large-area and high-efficiency displays.

## Methods

### Materials

Dimethyldichlorosilane (DMS, 99%) was purchased from Macklin. CdO (Cadmium oxide, 99.99%), ODE (1-octadecene, 90%), oleylamine (70%), sulfur (99.998%), ZnO (zinc oxide, 99.9%) and Zn(Ac)$_2$ (zinc acetate, 99.99%) were obtained from Sigma-Aldrich. OA (Oleic acid, 90%), TOP (trioctylphosphine, 90%), Se (selenium, 99.99%) were sourced from Alfa. TFB (poly(9,9-dioctylfluorene-co-N-(4-(3-methyl-propyl))-diphenylamine)) was sourced from ADS (American Dye Source), and PEDOT:PSS (4083) from Heraeus Deutschland GmbH & Co. KG. ITO substrates (10 Ω sq$^{-1}$) were supplied by Yiyang Huana

Xiangcheng Technology Co., Ltd. Red, green InP QDs and blue ZnSeTe QDs (25 mg mL$^{-1}$) were provided by Suzhou Xingshuo Nanotech Co., Ltd. Heptane (99%), n-octane (99%) and 2-propanol (99.5%) were sourced from Innochem, while high-purity n-hexane, ethanol, acetone, toluene and chlorobenzene were obtained from China National Pharmaceutical Group Corporation.

## Synthesis of Cd-based QDs

Preparation of precursor solutions. Synthesis of Zn(OA)$_2$ (zinc oleate) precursor: ZnO (15 mmol), OA (15 mL), and ODE (45 mL) were loaded into a 100 mL flask, and degassed for 30 min (at 150 °C), then heated under a nitrogen atmosphere to 300 °C with continuous stirring until a pale yellow. Synthesis of Cd(OA)$_2$ (cadmium oleate) precursor: Similarly, CdO (15 mmol), OA (15 mL), and ODE (45 mL) were combined in a flask (100 mL), and degassed for 30 min (at 150 °C), and then heated under nitrogen to 250 °C with continuous stirring until a slightly yellow (transparent sulotion). The resulting solution was cooled to 50 °C before use. Preparation of TOP-Se precursor: Se (30 mmol) was dispersed in TOP (120 mL) in a flask (250 mL), and stirred at 50 °C under nitrogen until a colorless, transparent solution was obtained. Preparation of TOP-S precursor: Sulfur powder (30 mmol) was similarly dissolved in TOP (120 mL) in other flask.

Growth of red CdSe/CdZnSe/ZnSeS/ZnS QDs. In a 100 mL flask, CdO (0.4 mmol), OA (1 mL), and ODE (15 mL) were mixed. Subsequently, heat the mixture to 150 °C under a flow of high-purity nitrogen and degas for 30 min. Then raise temperature to 310 °C. At this temperature, rapidly inject 3 mL of TOP-Se, maintain the temperature, and allow core growth for 10 min. Next, mix Zn(OA)$_2$ (3 mL), Cd(OA)$_2$ (3 mL), and TOP-Se (6 mL), and drip (6 ml h$^{-1}$) into the reaction solution. Then mix Zn(OA)$_2$ (3 mL), TOP-Se (1.5 mL), and TOP-S (1.5 mL), and similarly drip (6 mL h$^{-1}$). Finally, mix Zn(OA)$_2$ (5 mL) and TOP-S (5 mL), and continued to add dropwise at 6 mL h$^{-1}$. After completing reaction, cool the reaction solution to room temperature, purify it using n-hexane and acetone, and prepare a 20 mg mL$^{-1}$ octane solution.

Growth of Green CdSe/ZnSe/ZnSeS/ZnS QDs. To begin with, mixture CdO (0.2 mmol), Zn(Ac)$_2$ (4 mmol), OA (5 mL), and liquid paraffin (15 mL) was prepared in flask. Heat the solution to 130 °C under a flow of high-purity nitrogen and degas for 30 min. Next, heat to 330 °C, rapidly inject TOP-Se (2 mL), reduce temperature to 310 °C, and allow core growth for 5 min. Then, add 3 mL of a 1:1 Zn(OA)$_2$/TOP-Se mixed, and drip the mixture into flask at 6 mL h$^{-1}$. Next, combine Zn(OA)$_2$ (1.5 mL), TOP-Se (0.75 mL), and TOP-S (0.75 mL), and add dropwise at 0.1 mL min$^{-1}$. Finally, add 10 mL of a 1:1 Zn(OA)$_2$/TOP-S mixed at the same rate. After completing reaction, purify it using n-hexane and acetone, and prepare an 18 mg mL$^{-1}$ octane solution.

Growth of Blue CdZnSe/ZnSe/ZnSeS/ZnS QDs. In 100 mL flask, Zn(Ac)$_2$ (4 mmol), OA (4 mL), and ODE (10 mL) were degassed at 150 °C for 30 min, sealed and heated up to 300 °C. A mixture TOP-Se (2.5 mL) and Cd(OA)$_2$ (0.5 mL) were injected rapidly. The CdZnSe QDs were extracted by washing with ethanol. For core-shell QDs, heat the mixture (CdZnSe dispersion (2 mL), ODE (20 mL), OA (3 mL), and oleylamine (3 mL)) to 150 °C under a flow of high-purity nitrogen and degas for 30 min, raised to 300 °C. A mixture of TOP-Se (4 mL) and Zn(OA)$_2$ (4 mL) was added at 6 mL h$^{-1}$, followed by TOP-S/Se (2 mL) and Zn(OA)$_2$ (2 mL), and TOP-S (2 mL) and Zn(OA)$_2$ (4 mL), all at 6 mL h$^{-1}$. Washed three times, and prepare an 18 mg mL$^{-1}$ octane solution.

## Fabrication of uniform quantum dot films

The Shipley Microposit S1800 series photoresist was uniformly spin-coated onto a silicon wafer (with a diameter of 10 cm and a thickness of 525 μm) at speed 3000 rpm. Subsequently, ultraviolet exposure was performed using a photomask with a line width of 20 μm and a spacing of 1 μm. After development, the patterned wafer was etched by deep reactive ion etching (DRIE, Alcatel 601E) with fluorine-based gases. The

etching time was varied from 10 s to 6 min according to the target microstructure height. Upon completion of etching, the residual photoresist was removed using Microposit Remover 1165, followed by sequential cleaning with acetone and ethanol to eliminate organic residues. After treatment, the template was immediately immersed in a toluene solution containing DMS ($0.24 \times 10^{-3}$ M) (the water-saturated concentration in toluene being $C_{water} \approx 0.024 \times 10^{-3}$ M) for 1-2 min. After immersion, the template was sequentially rinsed with toluene and anhydrous ethanol. After rinsing, the template was oven-dried at 90 °C for 30 min. In a nitrogen atmosphere (inside a glovebox), the HTL (TFB) was coated onto the substrate via spin coating. Subsequently, the QD solution (8 mg mL$^{-1}$) was evenly dispensed onto the HTL surface using a pipette. The modified micropillar template was then placed over the QD solution to form a sandwich structure, and uniform pressure was applied on both sides. The volume of the QD solution was adjusted according to the film area to ensure that the sandwich structure was fully filled with the solution. Finally, the sandwich system was placed in a nitrogen atmosphere, where the solvent was allowed to completely evaporate, leading to the uniform and dense arrangement of QDs on the HTL layer, ready for subsequent device processing.

## Device fabrication

Synthesis of Zn$_{0.85}$Mg$_{0.15}$O nanoparticles (NPs). Zn$_{0.85}$Mg$_{0.15}$O NPs were synthesized using a solution precipitation method with modifications based on a literature procedure. Hydrated Zn(AC)$_2$ (5.1 mmol) and Mg(AC)$_2$ (0.9 mmol) were dissolved in DMSO (60 mL) and mixed with ethanolic (20 mL) TMAH solution (10 mmol), and stirred at 15-20 °C for 2 h. Subsequently, purification of the product was achieved through two consecutive dispersion/precipitation steps using ethanol/normal hexane. Finally, yielding high-purity Zn$_{0.85}$Mg$_{0.15}$O NPs.

Fabrication of QLEDs. ITO were sequentially treated with deionized water (30 min), acetone (900 s), isopropanol (900 s), and ozone (20 min). PEDOT:PSS was spin-coating ITO substrate at 5000 rpm, and subsequently annealed at 150 °C for 30 min. After PEDOT:PSS deposition, the substrates were transferred into an oxygen- and moisture-free glovebox, where TFB was spin-coated at 2000 rpm, annealing at 150 °C for 30 min on a hotplate. QD films were deposited onto the TFB layer either via evaporation-driven self-assembly (8 mg mL$^{-1}$, octane) or by conventional spin-coating (15 mg mL$^{-1}$, octane, 3000 rpm, 100 °C, 5 min). Afterward, a Zn$_{0.85}$Mg$_{0.15}$O nanoparticle layer (in anhydrous ethanol) was spin-coated at 3000 rpm and thermally treated on a hotplate at 60 °C for 30 min. Al (100 nm) was subsequently deposited via thermal evaporation.

Fabrication of patterned QLEDs. The fabrication process of patterned QLEDs is similar to that of conventional QLEDs. First, PEDOT:PSS and TFB layers are sequentially deposited onto the ITO substrate via spin-coating. Then, photoresist is spin-coated at a speed of 5000 rpm (30 s) and thermally treated at 100 °C for 120 s. The sample is subsequently transferred from the glovebox to a photo-lithography machine, where a patterned or pixelated mask is applied for 5 s of UV exposure. The sample is then transferred back to the glovebox, developed in hexane for 20 s, rinsed in ethyl acetate for 20 s, and thermally treated again at 180 °C for 300 s to form a patterned photoresist. Next, by employing the evaporation-driven QD self-assembly strategy, QDs are uniformly and densely deposited in the exposed areas of the photoresist pattern and on the TFB surface. Subsequently, a Zn$_{0.85}$Mg$_{0.15}$O nanoparticle layer and Al electrode are deposited, completing the fabrication of the patterned QLED device.

Fabrication of electron-only devices with different QD films. Electron-only devices (ITO/ZnMgO/QDs/ZnMgO/Al) were prepared by spin-coating a Zn$_{0.85}$Mg$_{0.15}$O layer (30 mg mL$^{-1}$, 3000 rpm, 60 °C, 30 min) onto cleaned ITO substrates in a nitrogen glove box. QD layers were deposited either by self-assembly (8 mg mL$^{-1}$, octane) or spin-

coating (15 mg mL$^{-1}$, octane, 3000 rpm, 80 °C, 3 min), followed by a second Zn$_{0.85}$Mg$_{0.15}$O coating. A 100 nm-thick aluminum was subsequently deposited via thermal evaporation.

Fabrication of Hole-Only Devices (ITO/PEDOT:PSS/TFB/QDs/MoO$_3$/Al) with Two Types of QD Films. Here, the deposition parameters for PEDOT:PSS, TFB, and QDs are consistent with those used in the QLED fabrication process. Finally, MoO$_3$ (10 nm) and Al (100 nm) are deposited by thermal evaporation.

## Characterizations
SEM images were acquired using a Hitachi SU8010 (Japan) at 20 kV and 15 μA beam current. TEM images was performed on JEOL TEM-2100F under an accelerating voltage of 200 kV. The fluorescence spectra, PL quantum efficiencies and PL decay dynamics were collected using transient/steady-state fluorescence spectrometer (Edinburgh Instruments). Absorption spectra were collected on a UV-visible spectrometer (Agilent Cary 7000). The GISAXS images were measured and collected using the Pilatus3R detector (Xeuss 2.0 HR) at an incident angle of 0.3°. All EL images were captured using a fluorescence microscope (OLYMPUS, IX 83, Japan) and own equipment. Contact angle measurements were performed using an OCA25 instrument (Germany). The roughness map of the film was acquired by AFM (Dimension FastscanBio, Bruker). The thermal image of the sample surface was acquired by Fluke Ti401 PRO camera. The capacitive-voltage (C-V) curve was tested by Phystech FT 1030 deep level transient spectrometer. The Current-voltage curves are measured by the Keithley 4200A-SCS system. A high-speed photomultiplier tube (Hamamatsu H10721-20) collected the tr-EL signal, converting it into a current.

## Light-emitting diodes characterizations
The photoelectronic performance parameters of all QLEDs, including EQE curves, operational lifetime curves, current density-voltage-luminance relationships, and EL spectral characteristics, were systematically characterized using a commercial fluorescence/EQE test system (Xi Pu Optoelectronics) within an inert nitrogen atmosphere glove box.

## Data availability
All data supporting the findings of this study are available within the paper and its Supplementary files. Any additional information related to the study is available from the corresponding author upon request. Source data are provided with this paper.

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

## Acknowledgements

H.L. and W.W. acknowledges financial support from the National Key Research and Development Program of China (no. 2024YFB3612800). Y.C.W. acknowledges funding support from the National Natural Science Foundation of China (nos. T2425026 and 52173190) and Youth Innovation Promotion Association CAS (no. 2018034). J.Z. acknowledges being supported by the Fundamental Research Program of Shanxi Province (no. 202303021222213). Y.Z. acknowledges the National Natural Science Foundation of China (52403310). H.G. acknowledges the National Natural Science Foundation of China (22205077).

## Author contributions

H.L., W.W. and Y.C.W. conceived and coordinated this work. B.J., Y.Z. and H.G. conducted a fluid dynamics analysis. H.L. and J.Z. fabricated the devices and collected the performance data of the LEDs. Y.J.W. provided InP QDs. Y.C.W., W.W., B.J. and H.L. drafted the paper with inputs from all authors. W.W., L.J. and Y.C.W. supervised the project.

## Competing interests

The authors declare no competing interests.
