## [Transparent Peer Review file · Nature Communications]

Highly Efficient Light-Emitting Diodes via Self-Assembled InP Quantum Dots

Corresponding Author: Professor Yuchen Wu

Version 0:

Reviewer comments:

Reviewer #1

(Remarks to the Author)

The manuscript is entitled "Highly Efficient Quantum Dot Light Emitting Diodes Achieved via Evaporation-Driven InP Quantum Dot Self-Assembly". My questions for this manuscript are as follows:

- 1) In Figure 3d, where is the blue emission coming from? The authors should describe the correlation between the roughness of the film and the blue emission.
- 2) In Figure 1d, the authors use SEM characterization and current density-voltage curves to show that films formed by evaporation-driven self-assembly are denser, have fewer defects, and exhibit less charge leakage compared to those formed by conventional spin coating. More evidence should be provided to support the claims of higher density and lower leakage current.
- 3) The article uses red InP quantum dots and mentions that the same assembly strategy applies to cadmium-based quantum dots. Is this assembly strategy not applicable to InP-based green or blue quantum dots? Why or why not? Further experimental results should be given.
- 4) On page 4, line 97, the authors describe the basic principle of evaporation-driven self-assembly. How the micropillar template anchors the liquid and the mechanisms for manipulating the TPCL?

Reviewer #2

(Remarks to the Author)

The manuscript "Highly Efficient Quantum Dot Light Emitting Diodes Achieved via Evaporation-Driven InP Quantum Dot Self-Assembly" by H. Li et al. demonstrated an evaporative-driven QD self-assembly strategy to enhance the density and uniformity of QD thin films, thereby significantly reducing charge leakage in QLED. QLEDs with dense and uniform InP-based QD films exhibit enhanced performance. Preparing high-quality QD thin films is an important topic, and the authors have proposed a feasible method. The following issues need to be addressed to improve the paper.

- (1) The formation of dense QD films is very important for high performance QLEDs. To fully and solidly verify that the density of the QD films is indeed improved, please measure and compare their refractive index, PL, absorption spectra, charge mobility and capacitance with those of spun-cast QD films of the same thickness.
- (2) Please compare the PL quenching of the dense QDs/ZnMgO with that of conventional QDs/ZnMgO. By improving the compactness of the QD films, the penetration of ZnMgO could be suppressed, leading to less emission quenching.
- (3) The enhanced performance is ascribed to the reduced leakage current by the dense and uniform QD films. However, according to Figure 3d, the reduction of TFB emission is not obvious. Please provide solid evidences to support that the leakage current is indeed reduced, and the enhanced performance is solely due to the reduced leakage current. The electrical modeling of the J-V and transient current curves can give an indication of the leakage current information.
- (4) The comparison in Figure 3h is not rigorous. To show that the reduction in device heat is due to a reduction in charge leakage, the authors should perform the comparison at the same current density. Likewise, Figure 3f should be plotted using EQE-J.
- (5) Supplementary Fig. 9 and Fig. 10, please compare the data with those of QLEDs fabricated by conventional spin-coating methods and see how much improvement can be achieved using the proposed method.
- (6) The stability of blue QLEDs is one of the limiting factors for the commercialization of QLEDs. As disclosed by many

papers, the electron leakage is one of the reasons for the low efficiency and short lifetime of blue QLEDs. The readers would be more interested to see the improvement effect (both efficiency and stability) of applying your method to blue QLEDs.

Could you please add these data?

(7) For display applications, the QD emitting layers should be patterned. Please state in the manuscript how to use your method to enhance the densities and uniformities of the patterned red, green and blue QD films. A demonstration would be better.

(8) Please detail the patterning process for the samples shown in Figure 4. Figure 4c is not discussed in the manuscript. Please specify the thickness of the patterned barrier layer, and if it is too thick, whether it will affect the smooth passage of the QD liquid.

(9) Provide additional details on the evaporation-driven self-assembly process to ensure reproducibility. A detailed preparation process should be provided in the Methods or Supplementary Materials.

(10) Please specify the solvents of all functional layers.

Reviewer #3

(Remarks to the Author)

Reviewer #4

(Remarks to the Author)

The authors demonstrated the process of obtaining micro-QLEDs. This research topic is potentially significant for the future of display devices. However, the current manuscript contains many unclear points. Therefore, I cannot recommend it in its current form. The major criticisms are listed below:

1) Poor Introduction Section: The authors emphasized that the evaporative-driven coating mechanism is crucial for achieving a uniform film. However, there is no literature survey on the evaporatively-driven coating method. The introduction is very poorly written.

For instance, there are well-known research works such as:

1. Marin, A.G., Gelderblom, H., Lohse, D., & Snoeijer, J.H. (2011). Order-to-disorder transition in ring-shaped colloidal stains. *Physical Review Letters*, 107(8), 085502.
2. Pyeon, J., Song, K.M., Jung, Y.S., & Kim, H. (2022). Self-Induced Solutal Marangoni Flows Realize Coffee-Ring-Less Quantum Dot Microarrays with Extensive Geometric Tunability and Scalability. *Advanced Science*, 9(11), 2104519.
3. Kim, H., Boulogne, F., Um, E., Jacobi, I., Button, E., & Stone, H.A. (2016). Controlled uniform coating from the interplay of Marangoni flows and surface-adsorbed macromolecules. *Physical Review Letters*, 116(12), 124501.
4. Park, Y., Lee, M., Seo, H., Shin, D., Hahm, D., Bae, W.K., ... & Kwak, J. (2024). Efficient and stable InP quantum-dot light-emitting diodes formed by premixing 2-hydroxyethyl methacrylate into ZnMgO. *Journal of Materials Chemistry C*, 12(20), 7270-7277.

2) Unclear Working Mechanism and Lack of Understanding of the Evaporative-Driven Coating Method: As the authors also addressed, a deeper understanding of the self-assembly mechanism to achieve a uniform film is necessary. However, this information is lacking. Most of the explanation is vague and lacks a clear picture.

Regarding the working mechanism, the authors should explain the following questions:

1. How strong is the disjoining pressure effect? Why does this force occur? What is the typical characteristic length? Have the authors considered the submicron film thickness or contact line structure? How do they know? If this is measured, how large is this force?
2. How do the capillary force and disjoining pressure force compare? Where do these forces act?
3. How strong is the evaporative flux, and how important is the evaporation effect? If the solvent is changed, are the results similar? Is the evaporation effect dominant here?
4. The evaporation mechanism is determined by diffusion, but air flow can create convection. What is the Peclet number in this case?
5. What about the effect of air flow on the top side of the meniscus?
6. How do the particles assemble without any defects? There is no explanation about particle-particle interactions.
7. How do the QD particles self-assemble into a uniform film shape?
8. There are no quantitative details to rationalize the coating mechanism.

3) Numerical Simulation Issues: In Figure 2b, the numerical simulation results are unclear. What is displayed in the figure? Is it force or stress? Is it related to the capillary force (or stress) or interfacial force (or stress) by the Marangoni effect? This is very unclear. Additionally, did the numerical simulation pass the mesh test? The contour plots seem unusual. The numerical simulation should be redone very carefully.

4) Scale-up Issue: There are several recent works on the fabrication of uniform QLED patterns. In this work, the authors suggest that the cover is very important. I am wondering if this method is feasible for mass production, considering the scale-up issue.

5) Minor Issues:

1. How did the authors obtain the color map in Supplementary Fig. 9? The measurement method was not fully described.
2. Why are the illuminated images blurred? Are there no sharp images? This is strange because the authors pointed out the

high resolution of the LED.

Reviewer #5

(Remarks to the Author)

Comments on Nature Communications manuscript

ID: NCOMMS-24-42981

Title: Highly Efficient Quantum Dot Light-Emitting Diodes Achieved via Evaporation-Driven InP Quantum Dot Self-Assembly

Authors: Hui Li, Jingyuan Zhang, Yuyan Zhao, Hanfei Gao, Jiangang Feng, Yunjun Wang, Lei Jiang, Yuchen Wu

This manuscript reports the fabrication of QLEDs with high electroluminescent performance as quantified by external quantum efficiency (EQE) and luminance. Self-assembled layers of quantum dots were used as uniform and dense films that formed in a solvent evaporation-driven process to replace the disordered, rough films previously used. The authors report reduced charge leakage and thermal losses. The use of superlattices as functional parts of QLEDs is certainly of interest to the community, and the manuscript is over all of good quality.

The assembly process that is driven by evaporation and guided by a microstructured surface is the main novelty of the manuscript. However, the process and its template are only described briefly, with few technical details. I do not believe that the contribution should be published unless detailed data on the geometries and deposition/evaporation conditions that yield the self-assembled films are added.

More specifically, I believe that the following points should be addressed:

(i) The statement on line 51: "Currently, the irregular morphology and non-uniform particle sizes of heavy-metal-free QDs present challenges for spin-coating to prepare dense and uniform films," is misleading. Uniform heavy metal-free QDs are state of the art (see, for example, Nat. Rev. Mater. 2023, 8, 742-758). It is unclear whether the remaining challenges are due to QD synthesis or their assembly via spin-coating. It is confusing that the authors themselves report in this contribution that InP particles (which, according to the authors, were uniform and regular in size), yielded poor QLEDs. If the assembly is the limiting step, the abovementioned statement should be replaced.

(ii) The evaporation-driven self-assembly is the most important process of this contribution, but few details are provided. The process is not described at all within "Methods". The geometry of the templates is only given as "width of 20 μm , an interval of 1 μm , and a 3 μm depression", without defining what "depressions" are, how the structures that supposedly stabilize the assembly are shaped, and how they work. It is unclear what the "stable environment is that the authors mention. It would be important to discuss whether the evaporation rate of the solvent play any role, whether it is possible to tune the thickness of the films, and how the uniformity of the films was quantified.

(iii) The authors selected red-emitting InP quantum dots but do not explain their choice. They state that "QD films prepared by the evaporation-driven strategy are denser and more uniform" (lines 168-169), but it is unclear what the reference is.

(iv) The authors report record-breaking EQE and luminance as summarized in Supplementary Table 1. A comparison directly in this table would aid the reader and underline the superior performance.

(v) The Methods section does not clearly state whether the protocols used in the preparation of InP QDs were novel or adapted from state of the art, and references to the base processes are missing. The same is true for the fabrication of the QLEDs.

The following two comments are suggestions and based on curiosity; the authors may choose whether to answer/consider them.

(vi) The stability of the fabricated QLEDs was analyzed. I was wondering whether the acidic PEDOT:PSS corrodes the ITO electrode.

(vii) Blue-emitting QDs (InP would be interesting because it does not contain Cd) are of great practical interest. Did the authors test whether their concept applies to them?

In summary, this contribution is of interest to readers of Nature Communication, but it lacks crucial information on a key process step and should not be accepted in this form. I believe that it will be possible to address this issue in a major revision.

Reviewer #6

(Remarks to the Author)

Version 1:

Reviewer comments:

Reviewer #1

(Remarks to the Author)

The authors have revised manuscript according to my comments.

Reviewer #2

(Remarks to the Author)

The authors have addressed most of my concerns and have made comprehensive revisions. Some of my concerns remain unaddressed.

1. The key point of the manuscript is the realization of dense QD films by evaporative-driven self-assembly strategy. This method can reduce the leakage current and potentially enhance the lifetime of QLED. However, as I mentioned in the first round of review, the leakage current is most significant in the blue QLED due to the difficulty of charge injection and poor charge balance. Therefore, the efficiency improvement in blue QLED should be the most significant, but I did not see these results in the manuscript. Besides, the authors should demonstrate the effects of leakage current reduction and lifetime enhancement in blue QLED, rather than in red devices.

2. Supplementary Fig. 18, the QLED with rough film exhibits higher EQE. Please double check.

3. I don't understand the purpose of demonstrating the patterned QLED (Fig. 4). As I pointed out in the first round of review, this method cannot be used to pattern the red, green, blue QD EML, and thus is useless for manufacturing full-colour QLED displays. To avoid misleading the readers, Fig. 4 and related discussion should be deleted.

Reviewer #3

(Remarks to the Author)

Reviewer #4

(Remarks to the Author)

Overall, the authors have taken the referee's critiques seriously and made commendable efforts to strengthen their manuscript. They have shifted the main emphasis toward materials and device applications, incorporated relevant literature, provided partial quantitative data, and revised figures and descriptions. These revisions clarify the core outcomes—specifically, the formation of quantum dot (QD) thin films and the enhancement of LED performance. While these changes do improve the manuscript's persuasiveness through the inclusion of performance metrics and strengthened references, there remains a significant need for a comprehensive explanation of the fluidic mechanisms involved, as well as a higher level of numerical rigor.

Despite the progress made in addressing Reviewer 4's concerns, the authors' claims exhibit notable inconsistencies. For instance, they describe particles as being influenced by long-range force interactions in one section, while suggesting in another that these forces can be disregarded. This contradiction undermines the coherence of their argument and detracts from the overall persuasiveness of the manuscript.

Moreover, the authors' treatment of evaporation, which is intrinsically linked to mass and phase transfer phenomena, suggests a lack of deep understanding in these areas. Given this, it may be more appropriate for this work to be submitted to a materials science or applied technology journal, rather than one that requires a robust grasp of the underlying fluid-mechanical or physical principles.

In summary, while the manuscript has improved in certain aspects, further refinement is necessary to address the mechanistic concerns raised and to ensure a consistent and coherent argument throughout.

Reviewer #5

(Remarks to the Author)

This report concerns the revised manuscript "Highly Efficient Light-Emitting Diodes via Self-Assembled InP Quantum Dots" that was submitted by Hui Li, YuchenWu, and co-authors for consideration at Nature Communications.

The revised manuscript and the detailed rebuttal letter addressed all concerns of the reviewers and resolved most. The only remaining concern is that the geometrical details of the micropillar template were only provided in the rebuttal (as Fig. R19) but not in the main manuscript or SI. We urge the authors to include this material for the readers.

The amended manuscript can be accepted for publication in our opinion.

Reviewer #6

(Remarks to the Author)

Version 2:

Reviewer comments:

Reviewer #2

(Remarks to the Author)

Although the authors have responded my questions, some of my concerns remain unaddressed.

For Fig. 4, I don't understand the motivation for your patterning method. By patterning the electrode, which is quite simple, you can achieve the same result, so what is the advantage of your method? It would be better if you could state your motivation and explain the advantage of your method. Also, the main focus of this paper is the formation of dense QD film and its effect on device performance. The patterning results are not so related to the main topic of the paper. To avoid misleading the reader, Figure 4 and the associated discussion should be deleted.

Reviewer #3

(Remarks to the Author)

Reviewer #4

(Remarks to the Author)

In the response regarding interparticle interactions, the authors explain that the influence of forces between quantum dots (QDs) varies with particle concentration—stating that at low concentrations, solvation forces dominate, while at high concentrations near the three-phase contact line, van der Waals interactions become significant. Although this explanation is fundamentally correct, the revised manuscript fails to explicitly acknowledge the ambiguity in the original draft. It would strengthen the response if the authors clearly admitted that the initial description was vague and then detailed how they have revised the explanation to resolve the inconsistency. This more transparent approach would help to clarify the rationale behind the concentration-dependent behavior of the interparticle interactions.

Regarding the issue raised by Reviewer 2 on the presentation of patterned QLEDs in Fig. 4, the reviewer noted that the demonstrated patterning method is unsuitable for fabricating full-color QLED displays, potentially misleading readers. In their response, the authors argued that the method holds intrinsic research value as a distinct technical pathway and therefore should be retained in the manuscript. However, this response does not fully address the reviewer's concern about possible confusion among readers. The authors should commit to including a clear statement—either in the main text or in the figure caption—that explicitly outlines the limitations of the patterning method, emphasizing that it is not applicable for full-color device manufacturing. Such a clarification would acknowledge the reviewer's point and mitigate potential misunderstandings about the method's applicability.

Response to Reviewers and Revised Details

Response to Reviewer 1:

Comment 1: The manuscript is entitled “Highly Efficient Quantum Dot Light Emitting Diodes Achieved via Evaporation-Driven InP Quantum Dot Self-Assembly”. My questions for this manuscript are as follows:

Response to Comment 1:

We appreciate Reviewer 1’s time, effort, and constructive comments on our manuscript. In response to Reviewer 1’s valuable suggestions, we have conducted additional experiments and revised the manuscript accordingly. We believe these revisions have significantly improved the manuscript and support its publication in *Nature Communications*.

Comment 2: In Figure 3d, where is the blue emission coming from? The authors should describe the correlation between the roughness of the film and the blue emission.

Response to Comment 2:

Thanks for raising this constructive feedback. The blue emission observed in Fig. 3d can be attributed to leakage currents stemming from the rough quantum dot (QD) film in the device and the subsequent radiative recombination occurring within the hole transport layer (HTL), as also observed in previous publications (*Nat. Photon.* **16**, 505–511 (2022); *Nat. Nanotechnol.* **18**, 1168–1174 (2023)). Especially, the rough QD film contains more surface defects, which may create additional leakage pathways, allowing some injected electrons to pass through the QD layer and reach the HTL directly. Given the wide bandgap of the HTL material, these electrons can undergo radiative recombination with holes in the HTL, resulting in the emission of blue light. In contrast, a uniform and dense QD film can effectively prevent electron injection into HTL, thereby avoiding the blue emission.

The surface roughness of the film significantly influences the charge transport and recombination processes within the device (*J. Am. Chem. Soc.* **140**, 8690-8695 (2018); <https://doi.org/10.1016/j.esci.2023.100227>). In rough QD films, the increased surface irregularities and defects are more likely to form leakage pathways, enabling some electrons to escape from the QD layer and directly enter the HTL. Conversely, a uniform and dense QD film can effectively block electron leakage, reduce non-radiative recombination, and thus almost entirely prevent blue emission.

In revised manuscript, we have included electroluminescence (EL) spectra of red InP-based QLEDs with varying uniformity and density, as shown in Fig. R1. As the uniformity and density of the QD films increase, the emission intensity in the blue

region gradually decreases and eventually disappears.

We have revised our manuscript as follows and supplemented related data in revised SI (Supplementary Fig. 14).

Fig. 3d presents a comparison of the EL spectra of QLEDs based on the two types of QD films at the same current density ($\sim 600 \text{ mA cm}^{-2}$). As the uniformity and density of the QD film increase, the parasitic emission intensity from the HTL gradually decreases and eventually disappears (Supplementary Fig. 14)²⁻³. (Page 8, Lines 12-15)

Fig. R1 | SEM images of red InP-based QD films with varying degrees of uniformity and density, and corresponding EL spectra of QLEDs. a, SEM image of a rough QD film. b, SEM image of a medium-density QD film. c, SEM image of a dense QD film. d, EL spectra of red InP-based QLEDs corresponding to films with different uniformities and densities. As the uniformity and density of the QD film increase, the emission intensity in the blue region gradually decreases and eventually disappears. All scale bar, 100 nm.

Comment 3: In Figure 1d, the authors use SEM characterization and current density-voltage curves to show that films formed by evaporation-driven self-assembly are denser, have fewer defects, and exhibit less charge leakage compared to those formed by conventional spin coating. More evidence should be provided to support the claims of higher density and lower leakage current.

Response to Comment 3:

We greatly appreciate this suggestion. We have measured the EL spectra, transient electroluminescence (TrEL), and capacitance-voltage curves of QLEDs based on two different thin films under identical current density conditions (*Nat. Photon.* **16**, 505–511 (2022); *Adv. Mater.* **35**, 2303950 (2023); *Nat. Commun.* **15**, 783 (2024); *Nat. Nanotechnol.* **18**, 1168–1174 (2023)), as shown in Fig. R2. At the same current density (approximately 600 mA cm^{-2}), rough QD films exhibited significant parasitic emission in the blue region, whereas those with dense and uniform QD films did not, indicating reduced charge leakage. Furthermore, QLEDs with dense and uniform QD films showed shorter turn-on times, faster TrEL responses, and lower capacitance across the entire voltage range. These findings suggest an enhanced electron-hole radiative recombination rate and reduced charge accumulation at the TFB/QD interface, effectively mitigating charge leakage.

We have revised our manuscript (Fig. 3d) as follows and supplemented related data in revised SI (Supplementary Fig. 15).

Fig. 3d presents a comparison of the EL spectra of QLEDs based on the two types of QD films at the same current density ($\sim 600 \text{ mA cm}^{-2}$). As the uniformity and density of the QD film increase, the parasitic emission intensity from the HTL gradually decreases and eventually disappears (Supplementary Fig. 14)²⁻³. (Page 8, Lines 12-15)

Additionally, we conducted transient electroluminescence (TrEL) measurements on QLEDs (Supplementary Fig. 15b). The comparison of TrEL kinetics reveals that QLEDs with uniform, dense QD films exhibit shorter turn-on times and faster TrEL. (Page 8, Lines 31-34)

To further understand the enhancement in performance, we measured the capacitance-voltage characteristics of QLED devices with both types of QD films (Supplementary Fig. 15a). The QLED with the uniform, dense QD film exhibits lower capacitance across the entire voltage range, indicating suppressed charge accumulation. (Page 8, Lines 27-31)

Fig. R2 | EL spectra, TrEL, and capacitance-voltage curves of QLEDs with two types of

red InP-based QD films (rough and uniform). **a**, The EL spectra were collected and normalized at the same current density (approximately 600 mA cm^{-2}). **b**, TrEL results. At 3.5 V, the QLED with the uniform, dense QD film shows shorter turn-on times and faster transient electroluminescence. These results indicate that the uniform, dense QD film enhances the electron-hole radiative recombination rate while significantly suppressing charge leakage. **c**, Capacitance-voltage characteristics. When the applied voltage exceeds 2 V, the capacitance of both devices increases, indicating charge carrier accumulation at the TFB/QD interface. As the voltage rises to 2.5 V, the capacitance begins to decrease, suggesting breakdown of the capacitance at the TFB/QD interface due to sufficient hole injection. The QLED with the uniform, dense QD film exhibits lower capacitance across the entire voltage range, indicating reduced charge accumulation.

Comment 4: The article uses red InP quantum dots and mentions that the same assembly strategy applies to cadmium-based quantum dots. Is this assembly strategy not applicable to InP-based green or blue quantum dots? Why or why not? Further experimental results should be given.

Response to Comment 4:

Thanks for this constructive suggestion. Exactly, evaporation-driven self-assembly strategy is also applicable to InP-based green or blue QDs. We integrated the InP-based green QDs and ZnSeTe-based blue QD films into QLED devices for performance testing, they exhibited similar trends to the InP-based red QLEDs, as shown in Fig. R3. QLEDs based on dense and uniform QD films demonstrated lower current density in the ohmic contact region. Like red counterparts, densely packed green and blue QDs also exhibit improved brightness and EQE compared to spin-coated QD films. However, due to the commercial InP and ZnSeTe QDs exhibit moderate quality, we cannot achieve state-of-the-art performances at the present stage (*Commun. Mater.* **2**, 96 (2021); *Adv. Funct. Mater.* **31**, 2008453 (2021); *Nanoscale* **15**, 2837-2842 (2023); *Adv. Mater.* **35**, 2303528 (2023)).

We have revised our manuscript as follows and supplemented related data in revised SI (Supplementary Fig. 18).

To confirm the versatility of the evaporation-driven self-assembly strategy, we fabricated QLEDs by self-assembly of InP-, ZnSeTe- and Cd-based QDs. All of these devices show significant enhancement of performances compared to the spin-coated films (Supplementary Figs. 17-18). (Page 11, Lines 10-13)

Fig. R3 | Uniform QD films prepared through the evaporation-driven self-assembly strategy resulted in enhanced electrical performance for green InP-based and blue ZnSeTe-based QLEDs. a-b, SEM images of green InP-based QD (a) and blue ZnSeTe-based QD (b) films prepared via the evaporation-driven self-assembly strategy. The images highlight the uniformity of the QD films, demonstrating the versatility of this self-assembly approach in fabricating high-quality QD films across different compositions. All scale bar, 100 nm. **c-d,** Current density-luminance-voltage (c) and EQE-current efficiency-current density (d) characteristics of green InP-based QLEDs. Compared to QLEDs fabricated using conventional methods, those with uniform QD films exhibit significantly lower current density in the ohmic contact region, with the EQE increasing from 7.2% to 9.8% and the luminance improving from $3.5 \times 10^4 \text{ cd m}^{-2}$ to $5.0 \times 10^4 \text{ cd m}^{-2}$. **e-f,** Current density-luminance-voltage (e) and EQE-current efficiency-current density (f) characteristics of blue ZnSeTe-based QLEDs. QLEDs with uniform QD films show a significant reduction in current density within the ohmic contact region, with the EQE rising from 6.7% to 8.6% and the luminance increasing from $2.0 \times 10^4 \text{ cd m}^{-2}$ to $2.3 \times 10^4 \text{ cd m}^{-2}$.

Comment 5: On page 4, line 97, the authors describe the basic principle of evaporation-

driven self-assembly. How the micropillar template anchors the liquid and the mechanisms for manipulating the TPCL?

Response to Comment 5:

We appreciate this valuable comment. As illustrated in Fig. R4a, we employed a lyophobic micropillar template to anchor the three-phase contact line (TPCL), which have been demonstrated in our previous publications (*Adv. Mater.* **28**, 3732 (2016); *Adv. Mater.* **30**, 1707291 (2018); *Nat. Commun.* **10**, 3912 (2019); *Adv. Mater.* **34**, 2202119 (2022)). In this configuration, evaporation of solvents near the TPCL sustains net capillary microfluid from liquid bulk to TPCL, which drives collective motion of QDs. Consequently, dense QD packing preferentially occurs at the TPCL for the generation of high-quality QD films.

To strengthen our manuscript, we have included following discussions in revised manuscript.

Influenced by the surface tension of the solution, the continuous liquid film exhibits a physical pinning effect on the micropillar template³⁶. (Page 6, Lines 2-4)

Response to Reviewer 2:

Comment 1: The manuscript “Highly Efficient Quantum Dot Light Emitting Diodes Achieved via Evaporation-Driven InP Quantum Dot Self-Assembly” by H. Li et al. demonstrated an evaporative-driven QD self-assembly strategy to enhance the density and uniformity of QD thin films, thereby significantly reducing charge leakage in QLED. QLEDs with dense and uniform InP-based QD films exhibit enhanced performance. Preparing high-quality QD thin films is an important topic, and the authors have proposed a feasible method. The following issues need to be addressed to improve the paper.

Response to Comment 1:

We sincerely appreciate Reviewer 2's high evaluations. As acknowledged by Reviewer 2, our manuscript presents a feasible approach to enhancing the density and uniformity of QD thin films, thereby significantly reducing charge leakage in QLED. In response to Reviewer 2's constructive feedback, we have made careful revisions to the manuscript. We believe these revisions have substantially improved the quality of our manuscript.

Comment 2: The formation of dense QD films is very important for high performance QLEDs. To fully and solidly verify that the density of the QD films is indeed improved, please measure and compare their refractive index, PL, absorption spectra, charge mobility and capacitance with those of spun-cast QD films of the same thickness.

Response to Comment 2:

We sincerely appreciate the valuable feedback from Reviewer 2. We have measured the refractive index, PL spectra, absorption spectra, charge mobility, and capacitance-voltage characterizations of two QD films with similar thicknesses. For approximately 20 nm thick QD films, the differences in optical properties, including refractive index, PL emission intensity, and absorption spectra, are unobservable. Then, we employed electrical measurements, including space-charge limited current (SCLC) and capacitance-voltage (C - V) measurements (Fig. R4). By fitting the J - V curves using the Mott-Gurney law, the charge mobility (μ can be calculated by):

$$J = \frac{9}{8} \varepsilon_0 \varepsilon_r \mu \frac{V^2}{L^3}$$

where ε_0 is the vacuum permittivity, $\varepsilon_r = 12.1$ is the relative permittivity of QDs (*Adv. Electron. Mater.* 2400142 (2024)), and L is the channel length. Higher electron (hole) mobility of $6.14 \times 10^{-4} \text{ cm}^2 \text{ V}^{-1} \text{ s}^{-1}$ ($5.21 \times 10^{-4} \text{ cm}^2 \text{ V}^{-1} \text{ s}^{-1}$) can be observed on evaporation-driven self-assembled QD films, while the spin-coated film exhibits a lower mobility of $4.39 \times 10^{-4} \text{ cm}^2 \text{ V}^{-1} \text{ s}^{-1}$ ($3.36 \times 10^{-4} \text{ cm}^2 \text{ V}^{-1} \text{ s}^{-1}$). The C - V curves

demonstrate that the QLEDs based on the evaporation-driven self-assembled QD films exhibit lower capacitance across the entire voltage range, indicating suppressed charge accumulation in densely packed self-assembled films. We have included additional relevant discussions in the revised manuscript.

The enhanced electron (Fig. 2g) and hole (Supplementary Fig. 11) mobilities (measured by space-charge limited current) and interface potential (Supplementary Fig. 8f) indicates that QD films prepared by the self-assembly exhibits more efficient carrier transport than spin-coated films, validating densely packed QD assembly. (Page 7, Lines 6-9)

To further understand the enhancement in performance, we measured the capacitance-voltage characteristics of QLED devices with both types of QD films (Supplementary Fig. 15a). The QLED with the uniform, dense QD film exhibits lower capacitance across the entire voltage range, indicating suppressed charge accumulation. (Page 8, Lines 27-31)

Fig. R4 | Comparison of electrical properties between QD films prepared by evaporation-driven self-assembly and spin-coating methods. a, The electron only device consist of ITO/ZnMgO/QDs/ZnMgO/Al layers, the holes are blocked by the ZnMgO layer. **b,** The SCLC

model is used to determine the electron mobility of homogeneous and dense QD films and coarse QD films. **c**, The hole only device consist of ITO/PEDOT:PSS/TFB/QDs/MoO₃/Al layers, the electrons are blocked by the MoO₃ layer. **d**, The hole mobility of homogeneous and dense QD films and coarse QD films. **e**, The ratio of electron mobility to hole mobility. **f**, Capacitance-voltage curves of QLED devices, indicating reduced charge accumulation for the QLED based on the evaporation-driven self-assembly QD film.

Comment 3: Please compare the PL quenching of the dense QDs/ZnMgO with that of conventional QDs/ZnMgO. By improving the compactness of the QD films, the penetration of ZnMgO could be suppressed, leading to less emission quenching.

Response to Comment 3:

Thanks for this constructive feedback. According to your suggestion, we measured PL emissions of the self-assembled and spin-coated QDs interfaced with ZnMgO. PL intensity of self-assembled QDs/ZnMgO is higher than that of spin-coated QDs/ZnMgO (Fig. R5a). To rationalize this PL enhancement, we further compare the time-resolved PL (Fig. R5b). The self-assembled QDs/ZnMgO exhibits a longer PL lifetime of 36.6 ns compared to 25.4 ns of its spin-coated counterpart, indicating suppressed ZnMgO penetration and PL quenching in self-assembled devices (*Adv. Mater.* 32, 2006178 (2020); *Adv. Optical Mater.* 11, 2300659 (2023)).

To address this point, we have added a relevant discussion in the revised manuscript.

Figs. 2e-f shows the time-resolved PL and steady-state PL spectra of self-assembled and spin-coated QDs interfaced with ZnMgO. Higher PL intensity, in line with longer PL lifetime, observed in self-assembled QDs indicates that the densely packed self-assembled QD films can effectively prevent ZnMgO penetration and PL quenching. (Page 7, Lines 2-6)

Fig. R5 | The steady-state fluorescence spectra and transient fluorescence decay dynamics of the two types of QD films on ZnMgO/QD/quartz structures. a, PL emission intensity comparison under identical excitation conditions. The PL intensity of the uniform and dense

QDs films is 18.6% higher than that of the rough QDs films. **b**, Time-resolved PL decay curves comparison. The PL lifetime of the uniform and dense QDs films is 36.6 ns, while that of the rough QDs films is 25.4 ns. These results indicate a reduction in non-radiative recombination processes in the uniform and dense films.

Comment 4: The enhanced performance is ascribed to the reduced leakage current by the dense and uniform QD films. However, according to Figure 3d, the reduction of TFB emission is not obvious. Please provide solid evidences to support that the leakage current is indeed reduced, and the enhanced performance is solely due to the reduced leakage current. The electrical modeling of the J - V and transient current curves can give an indication of the leakage current information.

Response to Comment 4:

We appreciate this constructive feedback. We agree with you that reduced TFB emission is not sufficient to demonstrate suppressed leakage current.

To further reveal the leakage current information, we conducted TrEL and C - V measurements (Fig. R6). Firstly, TrEL of self-assembled QD devices exhibits faster on-off speed, suggesting improved EL dynamics through suppressed leakage current. Secondly, C - V curves illustrate lower capacitance of self-assembled QD devices across the entire voltage range, indicating reduced charge accumulation at the TFB/QD interface.

About the transient current curves, unfortunately, we cannot conduct this measurement due to the technological limitation. However, we believe that the faster EL dynamics, together with reduced charge accumulation, can support dense packing of self-assembled QDs for suppressed leakage current.

In Fig. 1d of the original manuscript, we presented the J - V curves of the two QLEDs in the ohmic contact region. The results show that the current density of the QLED with the uniform QD film is an order of magnitude lower compared to the QLED with the rough QD film, indicating effective suppression of charge leakage (*Nat. Photon.* **16**, 505–511 (2022); *Nat. Nanotechnol.* **18**, 1168–1174 (2023)).

To address this issue, we have revised our manuscript as follows.

Fig. 3d presents a comparison of the EL spectra of QLEDs based on the two types of QD films at the same current density ($\sim 600 \text{ mA cm}^{-2}$). As the uniformity and density of the QD film increase, the parasitic emission intensity from the HTL gradually decreases and eventually disappears (Supplementary Fig. 14)²⁻³. (Page 8, Lines 12-15)

Additionally, we conducted transient electroluminescence (TrEL) measurements on QLEDs (Supplementary Fig. 15b). The comparison of TrEL kinetics reveals that

QLEDs with uniform, dense QD films exhibit shorter turn-on times and faster TrEL.
(Page 8, Lines 31-34)

To further understand the enhancement in performance, we measured the capacitance-voltage characteristics of QLED devices with both types of QD films (Supplementary Fig. 15a). The QLED with the uniform, dense QD film exhibits lower capacitance across the entire voltage range, indicating suppressed charge accumulation. (Page 8, Lines 27-31)

Fig. R6 | EL spectra, TrEL, and capacitance-voltage curves of QLEDs with two types of red InP-based QD films (rough and uniform). **a**, The EL spectra were collected and normalized at the same current density (approximately 600 mA cm⁻²). **b**, TrEL results. At 3.5 V, the QLED with the uniform, dense QD film shows shorter turn-on times and faster transient electroluminescence. These results indicate that the uniform, dense QD film enhances the electron-hole radiative recombination rate while significantly suppressing charge leakage. **c**, Capacitance-voltage characteristics. When the applied voltage exceeds 2 V, the capacitance of both devices increases, indicating charge carrier accumulation at the TFB/QD interface. As the voltage rises to 2.5 V, the capacitance begins to decrease, suggesting breakdown of the capacitance at the TFB/QD interface due to sufficient hole injection. The QLED with the uniform, dense QD film exhibits lower capacitance across the entire voltage range, indicating reduced charge accumulation.

Comment 5: The comparison in Figure 3h is not rigorous. To show that the reduction in device heat is due to a reduction in charge leakage, the authors should perform the comparison at the same current density. Likewise, Figure 3f should be plotted using EQE-J.

Response to Comment 5:

We sincerely appreciate the constructive feedback from Reviewer 2. In the revised manuscript, we have updated the data in Fig. 3h to compare the surface temperatures of the devices at the same current density, as shown in Fig. R7. Similarly, Fig. 3f has been redrawn using the EQE-current efficiency-current density relationship, as shown in Fig.

R8. The discussion in the manuscript has been revised as follows.

As the current density increases, we observed that the surface temperature of QLEDs with the rough QD film increased notably faster than those with the uniform films (Fig. 3h). At the current density of 700 mA cm^{-2} , the surface temperature of QLEDs with the rough QD film was $14.2 \text{ }^\circ\text{C}$ higher compared to those based on uniform films. (Page 10, Lines 11-15)

Fig. R7 | Evolution of surface temperature under different current densities for the two QLEDs. Insets shows infrared images of the two devices at current densities of 1 mA cm^{-2} and 700 mA cm^{-2} . When the current density increases to 700 mA cm^{-2} and stabilizes, the surface temperature of the QLED device using rough QD films rises to $43.1 \text{ }^\circ\text{C}$, while that of the QLED device using uniform QD films rises to $28.9 \text{ }^\circ\text{C}$.

Fig. R8 | EQE-current efficiency-current density characteristics for the two QLEDs. The QLED based on a uniform QD film exhibits higher EQE than the QLED with a rough QD film at the same current density. The peak EQE of the QLED with the uniform QD film reaches 26.6%, which is 1.4 times higher than the 18.9% peak EQE of the QLED with the rough QD film.

Comment 6: Supplementary Fig. 9 and Fig. 10, please compare the data with those of QLEDs fabricated by conventional spin-coating methods and see how much

improvement can be achieved using the proposed method.

Response to Comment 6:

We appreciate this valuable suggestion. We have compared the performances of Cd-based spin-coated and self-assembled QD devices. The results indicate that the current density-luminance-voltage characteristics exhibit similar trends to those observed in red InP-based QLEDs, as shown in Fig. R9c-e. The peak EQE shows enhancement from 21.6% to 29.6% for red QDs, from 19.3% to 25.3% for green QDs, and from 18.7% to 22.2% for blue QDs (Fig. R9a).

We have included related discussions in the revised manuscript.

To confirm the versatility of the evaporation-driven self-assembly strategy, we fabricated QLEDs by self-assembly of InP-, ZnSeTe- and Cd-based QDs. All of these devices shows significant enhancement of performances compared to the spin-coated films (Supplementary Figs. 17-18). (Page 11, Lines 10-13)

Fig. R9 | EQE, current density-luminance-voltage characteristics, and color space coordinates for red, green, and blue Cd-based QLEDs using two types of QD films. a, Compared to the maximum EQEs of 21.6%, 19.3%, and 18.7% for red, green, and blue QLEDs based on rough QD films, the maximum EQEs for QLEDs based on dense QD films are significantly improved to 29.6%, 25.3%, and 22.2%, respectively. This is one of the best results reported so far. **b,** CIE 1932 color space coordinates corresponding to the red, green, and blue Cd-based QLEDs in (a), with specific CIE coordinates of (0.684, 0.316), (0.227, 0.734), and (0.133, 0.060). The yellow triangle represents the DCI-P3 color space. These color gamuts surpass the Digital Cinema Initiatives P3 (DCI-P3) color space, a standard for wide-gamut

displays. **c-e**, Devices based on dense QD films exhibit lower current density in the ohmic contact region compared to those based on rough QD films, indicating effective suppression of leakage current. Additionally, devices with dense QD films show enhanced luminance: red Cd-based QLEDs (c) increased from 2.1×10^5 to 3.6×10^5 cd m^{-2} , green Cd-based QLEDs (d) from 3.8×10^5 to 5.6×10^5 cd m^{-2} , and blue Cd-based QLEDs (e) from 3.6×10^4 to 4.0×10^4 cd m^{-2} , compared to devices with rough QD films.

Comment 7: The stability of blue QLEDs is one of the limiting factors for the commercialization of QLEDs. As disclosed by many papers, the electron leakage is one of the reasons for the low efficiency and short lifetime of blue QLEDs. The readers would be more interested to see the improvement effect (both efficiency and stability) of applying your method to blue QLEDs. Could you please add these data?

Response to Comment 7:

We agree with Reviewer 2 that the applications of our method in blue QLEDs will highlight the importance of our manuscript. According to your constructive feedback, we have supplemented LED performances of ZnSeTe and Cd-based QDs, indicating enhanced EQE in self-assembled QD devices. Regarding stability, we have conducted operational lifetime measurements for Cd-based blue QLEDs, as shown in Fig. R10e-f. Based on the initial brightness of $1,000 \text{ cd m}^{-2}$, we obtain a T_{95} of 19 hours in self-assembled QDs.

Fig. R10 | Uniform QD films prepared through the evaporation-driven self-assembly strategy resulted in enhanced performance for blue ZnSeTe-based and Cd-based QLEDs.

a-b, Current density-luminance-voltage (a) and EQE-current efficiency-current density (b) characteristics of blue ZnSeTe-based QLEDs. QLEDs with uniform QD films show a significant reduction in current density within the ohmic contact region, with the EQE rising from 6.7% to 8.6% and the luminance increasing from 2.0×10^4 cd m⁻² to 2.3×10^4 cd m⁻². **c-d**, Current density-luminance-voltage (c) and EQE-current efficiency-current density (d) characteristics of blue Cd-based QLEDs. QLEDs with uniform QD films show a significant reduction in current density within the ohmic contact region, with the EQE rising from 18.7% to 8.6% and the luminance increasing from 2.0×10^4 cd m⁻² to 2.3×10^4 cd m⁻². **e**, Relative luminance-time characteristics of Cd-based blue QLED. **f**, T_{95} lifetime measurements of Cd-based blue QLED at different initial luminance.

Comment 8: For display applications, the QD emitting layers should be patterned. Please state in the manuscript how to use your method to enhance the densities and uniformities of the patterned red, green and blue QD films. A demonstration would be better. Please detail the patterning process for the samples shown in Figure 4. Figure 4c is not discussed in the manuscript. Please specify the thickness of the patterned barrier layer, and if it is too thick, whether it will affect the smooth passage of the QD liquid.

Response to Comment 8:

We agree with Reviewer 2. In our study, we adopted a charge-blocking QD layer patterning approach (*Nat. Photon.* 16, 297–303 (2022)). The specific process is shown in Fig. R11. In the patterned QLEDs, the blocking layer has a thickness of approximately 20 nm, as shown in Fig. R12. This thickness effectively suppresses charge leakage while ensuring that it does not disrupt the uniform deposition of the QD film under the evaporation-driven self-assembly method. In the revised manuscript, we have added a detailed discussion of Fig. 4c and supplemented the methods section with the preparation process of the patterned QLEDs. The detailed revisions are as follows.

These micropores or patterns were fabricated using photolithography, with the blocking layer thickness of 20 nm, eliminating the effect on TPCL slip (supplementary Fig. 19-20). (Page 11, Lines 25-27)

Fabrication of patterned QLEDs. The fabrication process of patterned QLEDs is similar to that of conventional QLEDs. First, PEDOT:PSS and TFB layers are sequentially deposited onto the ITO substrate via spin-coating. Then, photoresist is spin-coated at a speed of 4000 rpm for 30 s and thermally treated at 90 °C for 90 s. The sample is subsequently transferred from the glovebox to a photolithography machine, where a patterned or pixelated mask is applied for 5 s of UV exposure. The sample is then transferred back to the glovebox, developed in hexane for 20 s, rinsed in ethyl acetate for 20 s, and thermally treated again at 150 °C for 5 min to form a patterned photoresist layer. Next, by employing the evaporation-driven QD self-assembly strategy, QDs are

uniformly and densely deposited in the exposed areas of the photoresist pattern and on the TFB surface. Subsequently, a $Zn_{0.85}Mg_{0.15}O$ nanoparticle layer and Al electrode are deposited, completing the fabrication of the patterned QLED device. (Page 14, Lines 27-36; Page 15, Lines 1-2)

Fig. R11 | Schematic diagram of preparation process of patterned QLED.

Fig. R12 | Cross-sectional TEM image of patterned QLED. Scale bar, 100 nm.

Comment 10: Provide additional details on the evaporation-driven self-assembly process to ensure reproducibility. A detailed preparation process should be provided in the Methods or Supplementary Materials.

Response to Comment 10:

We appreciate Reviewer 2's feedback. We have provided a detailed explanation of the evaporation-driven self-assembly process used to fabricate uniform and dense QD films in the Methods section of the revised manuscript. The specific revisions are as follows:

Fabrication of uniform and dense quantum dot films.

First, the micropillar template was fabricated following the method reported in the literature^{27,32}. The micropillar template was then treated in a plasma cleaner for 15 min to remove surface impurities. After treatment, the template was immediately immersed in a toluene solution containing DMS (0.24×10^{-3} M) (the water-saturated concentration in toluene being $C_{\text{water}} \approx 0.024 \times 10^{-3}$ M) for 1-2 min. After immersion, the template was sequentially rinsed with toluene and anhydrous ethanol. After rinsing, the template was dried in an oven at 90 °C for 30 min. In a nitrogen atmosphere (inside a glovebox), the HTL (TFB) was coated onto the substrate via spin coating. Subsequently, the QD solution (8 mg mL^{-1}) was evenly dispensed onto the HTL surface using a pipette. The modified micropillar template was then placed over the QD solution to form a sandwich structure, and uniform pressure was applied on both sides. The volume of the QD solution was adjusted according to the film area to ensure that the sandwich structure was fully filled with the solution. Finally, the sandwich system was placed in a nitrogen atmosphere, where the solvent was allowed to completely evaporate, leading to the uniform and dense arrangement of QDs on the HTL layer, ready for subsequent device processing. (Page 13, Lines 16-31)

Comment 11: Please specify the solvents of all functional layers.

Response to Comment 11:

We would like to thank Reviewer 2 for their comments. We have provided detailed information regarding the solvents used for each functional layer in the QLED devices. The solvent for the emission layer (QDs) is n-octane. The solvent for the hole transport layer (TFB) is chlorobenzene. The solvent for the hole injection layer (PEDOT:PSS) is water. The solvent for the electron transport layer ($\text{Zn}_{0.85}\text{Mg}_{0.15}\text{O}$) is anhydrous ethanol. The cathode is deposited from aluminum in a vacuum environment.

To clarify the solvents used for each functional layer, we have made the following revisions to the manuscript.

Fabrication of QLEDs. The fabrication process of QLEDs comprises the following steps: Initially, a thorough cleaning of the indium tin oxide (ITO) glass substrate is conducted using a sequence of continuous cleaning processes involving deionized water for 30 min, acetone for 15 min, isopropanol for 15 min, and ozone treatment for 20 min. Subsequently, PEDOT:PSS (CLEVIOS P VP AI 4083; solvent: water) is uniformly spin-coated onto the ITO glass substrate at a speed of 4000 rpm, followed by a 30 min baking at 150 °C in ambient air. Next, the ITO substrate coated with PEDOT:PSS is transferred into a nitrogen-filled glovebox, where a TFB solution with a concentration of 8 mg mL^{-1} (solvent: chlorobenzene) is spin-coated onto the

PEDOT:PSS layer at 2000 rpm. This is followed by a 20 min baking step at 130 °C under a nitrogen atmosphere to form a uniform TFB layer. Within the glovebox, a specialized QD assembly technique is employed to assemble QDs (with a concentration of 8 mg mL⁻¹; solvent: octane) onto the TFB layer. In conventional QLEDs, the emissive layer is typically formed by spin-coating a QD solution with a concentration of 15 mg mL⁻¹ (solvent: octane) onto the TFB layer at 3000 rpm, followed by a 5 min baking step at 100 °C. Subsequently, Zn_{0.85}Mg_{0.15}O NPs (30 mg mL⁻¹; solvent: anhydrous ethanol) are spin-coated onto the QD layer at 3000 rpm, followed by a 30 min baking at 60 °C to create the NPs layer. Finally, the multi-layer structure is placed into a dedicated high-vacuum thermal evaporation chamber, where a patterned Al cathode layer with a thickness of 100 nm is deposited via in-situ masking, resulting in a QLED with an effective emissive area of 3 mm². (Page 14, Lines 7-26)

Response to Reviewer 3:

Comment 1: I co-reviewed this manuscript with one of the reviewers who provided the listed reports. This is part of the *Nature Communications* initiative to facilitate training in peer review and to provide appropriate recognition for Early Career Researchers who co-review manuscripts.

Response to Comment 1:

Thank you for your thorough review and valuable feedback on this manuscript. We are pleased to learn that you co-reviewed the manuscript with another reviewer, particularly in the context of *Nature Communications* initiative to involve Early Career Researchers in the peer review process. This initiative not only helps to enhance the overall quality of research but also provides invaluable learning and development opportunities for emerging scholars. We believe that such collaborative review processes allow for a more comprehensive evaluation of the manuscript's quality and further promote the growth and exchange within the research community. We sincerely appreciate the efforts you and your co-reviewer have dedicated to this review.

Response to Reviewer 4:

Comment 1: The authors demonstrated the process of obtaining micro-QLEDs. This research topic is potentially significant for the future of display devices. However, the current manuscript contains many unclear points. Therefore, I cannot recommend it in its current form. The major criticisms are listed below:

Response to Comment 1:

We sincerely thank Reviewer 4 for the time, effort, and constructive feedback on our manuscript. We acknowledge the major concerns raised regarding the "many unclear points" in the evaporation-driven assembly mechanism.

First, we would like to emphasize that **the main novelty and innovation of this work lie in the self-assembly fabrication of dense and uniform InP quantum-dot (QD) thin films for efficient LEDs**, as acknowledged by Reviewers 1 and 2. The dense QD packing achieved through this self-assembly method has enabled suppressed leakage current and reduced heat generation, leading to record-high LED performance in both external quantum efficiency and maximum luminance (see details in Supplementary Table 1). Moreover, these dense and flat self-assembled QD films have been successfully integrated into a micro-LED device configuration, showcasing their potential applications in near-eye display technology (see Fig. 4 for details). Therefore, the focus of this work is primarily on material fabrication and device applications rather than the microfluidic mechanisms involved in the self-assembly process.

Second, we fully acknowledge Reviewer 4's observation that the evaporation-driven self-assembly mechanism remains unclear in several aspects, such as the strength of particle-particle interactions and capillary forces. This limitation arises from our expertise being rooted in material science and optoelectronic devices, rather than in soft-matter physics or microfluid dynamics. Despite our limited background in these areas, we have made significant efforts and done our best to address the constructive comments raised by Reviewer 4, as detailed in our point-by-point response to each comment. To further clarify the scope of this manuscript and avoid potential misunderstandings, we have made the following major revisions to strengthen the focus of the work:

(1) The title has been revised from "*Highly Efficient Quantum Dot LEDs Achieved via Evaporation-Driven InP Quantum Dot Self-Assembly*" to "*Highly Efficient Light-Emitting Diodes via Self-Assembled InP Quantum Dots*".

(2) We have restructured Fig. 2, removing the panels related to microfluidics and introducing new panels focused on material and electrical characterizations. These additions aim to better illustrate the structure-performance relationship (see details in

the revised Fig. 2, Page 7).

We believe that this manuscript has been significantly improved through this round of revisions and is now suitable for publication in *Nature Communications*.

Comment 2: Poor Introduction Section: The authors emphasized that the evaporative-driven coating mechanism is crucial for achieving a uniform film. However, there is no literature survey on the evaporatively-driven coating method. The introduction is very poorly written. For instance, there are well-known research works such as:

1. Marin, A.G., Gelderblom, H., Lohse, D., & Snoeijer, J.H. (2011). Order-to-disorder transition in ring-shaped colloidal stains. *Physical Review Letters*, 107(8), 085502.
2. Pyeon, J., Song, K.M., Jung, Y.S., & Kim, H. (2022). Self-Induced Solutal Marangoni Flows Realize Coffee-Ring-Less Quantum Dot Microarrays with Extensive Geometric Tunability and Scalability. *Advanced Science*, 9(11), 2104519.
3. Kim, H., Boulogne, F., Um, E., Jacobi, I., Button, E., & Stone, H.A. (2016). Controlled uniform coating from the interplay of Marangoni flows and surface-adsorbed macromolecules. *Physical Review Letters*, 116(12), 124501.
4. Park, Y., Lee, M., Seo, H., Shin, D., Hahm, D., Bae, W.K., ... & Kwak, J. (2024). Efficient and stable InP quantum-dot light-emitting diodes formed by premixing 2-hydroxyethyl methacrylate into ZnMgO. *Journal of Materials Chemistry C*, 12(20), 7270-7277.

Response to Comment 2:

We agree with Reviewer 4 that a thorough survey on the evaporation-driven coating method is absent in our original Introduction section and we also thank you for recommending these literatures, which have been included in our revised manuscript.

To strengthen our manuscript, we have included related discussion as:

Various techniques have significantly improved the external quantum efficiency (EQE) of heavy-metal-free QLEDs^{9,13-16}, including using hydrofluoric acid treatment to suppress trap states caused by surface oxygen on the core, developing core-shell structures to passivate traps, employing ligands to passivate surface defects, and optimizing charge transport layers for better energy level alignment. (Page 3, Lines 8-12)

While multicomponent inks can suppress intense Marangoni flows by lowering the solvent evaporation rate, challenges remain in achieving dense, uniform, and ordered nanoparticle films as the evaporation rate increases²³⁻²⁵. The addition of surfactants is an effective method for achieving uniform films, but it introduces uncertainties in

device performance²⁶. Langmuir-Blodgett technique can achieve both ordered assembly of QDs and the preparation of large-area films, but its compatibility with device integration is limited²⁷. (Page 3, Lines 21-26)

The evaporation of the solvent at the liquid-gas interface creates a surface tension gradient that drives Marangoni flow towards the TPCL for mass transfer (as shown in Fig. 2b). A uniform flow region on the substrate side facilitates even mass transfer (Supplementary Fig. 2a)³⁸⁻³⁹. (Page 6, Lines 14-17)

16. Park Y, et al. Efficient and stable InP quantum-dot light-emitting diodes formed by premixing 2-hydroxyethyl methacrylate into ZnMgO. *J. Mater. Chem. C*. **12**, 7270-7277 (2024). (Page 16)

25. Marín, Á, G. et al. Order-to-Disorder Transition in Ring-Shaped Colloidal Stains. *Phys. Rev. Lett.* **107**, 085502 (2011). (Page 17)

26. Kim H. et al. Controlled Uniform Coating from the Interplay of Marangoni Flows and Surface-Adsorbed Macromolecules. *Phys. Rev. Lett.* **116**, 124501 (2016). (Page 17)

38. Pyeon J. et al. Self-Induced Solutal Marangoni Flows Realize Coffee-Ring-Less Quantum Dot Microarrays with Extensive Geometric Tunability and Scalability. *Adv. Sci.* **9**, 2104519 (2022). (Page 18)

Comment 3: Unclear Working Mechanism and Lack of Understanding of the Evaporative-Driven Coating Method: As the authors also addressed, a deeper understanding of the self-assembly mechanism to achieve a uniform film is necessary. However, this information is lacking. Most of the explanation is vague and lacks a clear picture. Regarding the working mechanism, the authors should explain the following questions: How strong is the disjoining pressure effect? Why does this force occur? What is the typical characteristic length? Have the authors considered the submicron film thickness or contact line structure? How do they know? If this is measured, how large is this force? How do the capillary force and disjoining pressure force compare? Where do these forces act?

Response to Comment 3:

Thank you for this feedback. Although we have limited expertise in microfluids, we have made significant efforts to understand and respond to these constructive comments. In this case, the disjoining pressure stems from the van der Waals interaction between solvent molecules and the QD ligands. We neglect the electrostatic interactions between particles considering charge neutrality of QDs. The magnitude of the disjoining pressure is primarily influenced by the height of the wedge-shaped liquid film (*Nature* **423**, 156–159 (2003)) and its characteristic length is in sub-micrometer

scale, which cannot be observed directly. According to the Derjaguin-Landau-Verwey-Overbeek (DLVO) theory (Langmuir, 39, 13359-13370 (2023)), we estimated the disjoining pressure as,

$$P = \frac{\delta}{6\pi h^3}$$

where δ represents the Hamaker constant of the n-octane solvent (i.e., the dispersive force), and h is the height of the wedge-shaped liquid film. The disjoining pressure is calculated as between 100 and 1000 Pa.

In confined capillary liquids, the capillary forces are primarily governed by Laplace pressure arising from the formation of the meniscus (Adv. Mater. 34, 2110695 (2022)). In our scenario, capillary forces can avoid the pinning of three-phase contact line, yielding continuous dewetting for dense and uniform QD packing. In contrast, the disjoining pressure primarily acts at the interface between the solvent and the QDs and their ligands, influencing the arrangement of the quantum dots and the structure of the self-assembled film.

To more accurately express these concepts and optimize our manuscript, we have revised the relevant statements in the revised version of our paper.

To mitigate the self-spreading behavior of octane at the edges, we designed a reverse-curved structure at both ends of the micropillar template (Supplementary Fig. 2b)³⁷. (Page 6, Lines 6-8)

Fig. R13 | Capillary force generated by the reverse-bent micropillars balances the self-spreading of octane.

Comment 4: How strong is the evaporative flux, and how important is the evaporation effect? If the solvent is changed, are the results similar? Is the evaporation effect dominant here? The evaporation mechanism is determined by diffusion, but air flow

can create convection. What is the Peclet number in this case? What about the effect of air flow on the top side of the meniscus?

Response to Comment 4:

We appreciate this comment. To evaluate the evaporation flux (J), we directly observed the receding of three-phase contact line in our self-assembly system (Fig. R14) and J can be expressed as,

$$J = \frac{\Delta V}{A\Delta t}$$

where ΔV is the evaporated volume, A is the area of the evaporation interface, and Δt is the time required for the evaporation volume (ΔV). The evaporated volume (ΔV) can be expressed by the following relation:

$$\Delta V = A \cdot \Delta l$$

Δl is the distance of the TPCL movement (which is consistent with the translation distance of the evaporation interface). Over a period of 715 s, the TPCL moved 331 μm . Therefore, the volumetric evaporative flux is given by $J = 4.63 \times 10^{-7} \text{ m}^{-1} \text{ s}^{-1}$. In the ideal case, where the evaporated solvent is fully converted into vapor, the flow velocity (v) can be calculated using the following formula.

$$v = \frac{\Delta V \rho * 22.4 \times 10^{-3}}{M\Delta t}$$

namely

$$v = \frac{\Delta l \rho * 22.4 \times 10^{-3}}{M\Delta t}$$

ρ is the density of octane, and M is the molecular weight of octane, which gives a v of $6.34 \times 10^{-5} \text{ m s}^{-1}$. According to physical handbooks, the diffusion coefficient (D) of octane vapor in air is approximately $0.24 \times 10^{-5} \text{ m}^2 \text{ s}^{-1}$. Using this, the Peclet number (P_e) can be calculated (*J. Math. Sci.* **253**, 168–179 (2021)).

$$P_e = \frac{vH}{D}$$

v represents the diffusion velocity of solvent molecules during evaporation, and H denotes the thickness of the liquid bridge. The P_e is significantly less than 1, indicating that the diffusion mechanism dominates the evaporation process, with the influence of convection being relatively small. Although a slight air flow may have a potential effect on the top of the meniscus, its impact is minimal and can be considered negligible.

Additionally, we replaced the solvent in the QD solution (including n-hexane,

cyclohexane, and toluene) and repeated the film preparation experiments (Fig. R15). The results indicate that the QD films prepared with these solvents exhibit comparable uniformity and density to those prepared with n-octane, further validating the universality of the self-assembly strategy.

We have revised our manuscript as follows and supplemented related data in revised SI (Supplementary Note 1 and Supplementary Figs. 1 and 10).

At temperatures of 25°C or lower, the solvent molecules are mainly diffused, and we can observe that the roughness of the prepared QD films is lower. The thickness of the film can be adjusted by varying the concentration of the QD solution (supplementary Fig. 4). (Page 6, Lines 17-20)

Fig. R14 | In situ fluorescence microscopy image showing the slip process of the TPCL. All scale bar, 100 μm.

Fig. R15 | Comparison of QD films prepared with different solvents. The image shows the films prepared using n-hexane, cyclohexane, and toluene alongside those prepared with n-octane. The results reveal that QD films from all solvents exhibit comparable uniformity and density, demonstrating the versatility of the evaporation-driven self-assembly strategy across different solvent systems. All scale bar, 100 nm.

Comment 5: How do the particles assemble without any defects? There is no explanation about particle-particle interactions. How do the QD particles self-assemble into a uniform film shape?

Response to Comment 5:

We sincerely appreciate the detailed comments from the reviewers. In the QD self-assembly process, the self-assembly is primarily driven by van der Waals forces and solvent evaporation dynamics. Initially, QDs remain well-dispersed in the solvent, with their stability primarily relying on solvation forces. However, as the solvent gradually

evaporates, the solvation forces weaken and can no longer maintain the stable dispersion of the QDs (*J. Chem. Phys.* **120**, 1921–1934 (2004), *J. Chem. Phys.* **120**, 9703–9714 (2004)). During this process, the molecular interactions between the QD ligands begin to dominate the self-assembly behavior (*Small* **5**, 1600–1630 (2009)). As the TPCL (the wedge-shaped liquid film) shifts, the QDs begin to precipitate and self-assemble, ultimately forming a uniform and dense QD film.

To enhance the quality of the manuscript, we have made the following revisions:

Compared to spin-coating deposition, our approach provides a slower and more controlled solvent evaporation rate, while suppressing disordered liquid flows. The forces between ligands contribute to the uniformity and density of QD film^{25,33-34}. (Page 4, Lines 28-30)

Comment 6: There are no quantitative details to rationalize the coating mechanism.

Response to Comment 6:

We sincerely appreciate the valuable comments provided by Reviewer 4. In self-assembly strategy, the influence of the template on the behavior of the liquid film, particularly the solvent evaporation rate and the stability of the TPCL, plays a crucial role. Through fluid dynamics simulations and experimental data, we observed that under the anchoring effect of the micropillar template, the liquid film forms a stable linear TPCL. Additionally, the semi-closed assembly system significantly reduces the solvent evaporation rate. After the solvent molecules evaporate, the Péclet number is much less than 1, indicating that the solvent molecules primarily diffuse, and thus, the effect on the curvature of the liquid interface can be neglected. As the wedge-shaped liquid film slides along the substrate surface, intermolecular forces drive the self-assembly of the QDs in the direction of the slide. Furthermore, by adjusting the temperature of the assembly system and the concentration of the QD solution, we can quantitatively control the film thickness and uniformity (Fig. R16). SEM, AFM, and GISAXS results provide quantitative data on the film's structure, roughness, and uniformity.

We have revised our manuscript as follows and supplemented related data in revised SI (Supplementary Figs. 4 and 6).

Compared to the case without micropillars, the TPCL on the substrate side transitions from a curved configuration to a linear configuration under the influence of capillary forces (Supplementary Fig. 2a). To mitigate the self-spreading behavior of octane at the edges, we designed a reverse-curved structure at both ends of the micropillar template (Supplementary Fig. 2b)³⁷. We also observed the jumping of the TPCL between the

micropillars, resulting in fluctuations on the substrate side. To better describe the impact of these fluctuations on QD self-assembly, we introduced the concept of error rate, defined as the ratio of fluctuation time to the total dewetting time within one cycle. Based on regions with an error rate below 5% (Supplementary Fig. 2c), we designed a micropillar template with a width of 20 μm , a spacing of 1 μm , and a depth of 3 μm for QD self-assembly (Supplementary Fig. 3). The evaporation of the solvent at the liquid-gas interface creates a surface tension gradient that drives Marangoni flow towards the TPCL for mass transfer (as shown in Fig. 2b). A uniform flow region on the substrate side facilitates even mass transfer (Supplementary Fig. 2a)³⁸⁻³⁹. At temperatures of 25°C or lower, the solvent molecules are mainly diffused, and we can observe that the roughness of the prepared QD films is lower. The thickness of the film can be adjusted by varying the concentration of the QD solution (supplementary Fig. 4). (Page 6, Lines 4-20)

Fig. R16 | The influence of temperature (solution concentration) on the roughness (thickness) of the QD film. **a**, Relationship curve between ambient temperature and roughness of QD films. At 25°C and below, the surface roughness of QD film is low, and the surface roughness increases significantly with the increase of temperature. **b**, The relationship between the thickness of the QD film and the concentration of QD solution. The film thickness increases almost linearly with the concentration of QD solution.

Comment 7: Numerical Simulation Issues: In Figure 2b, the numerical simulation results are unclear. What is displayed in the figure? Is it force or stress? Is it related to the capillary force (or stress) or interfacial force (or stress) by the Marangoni effect? This is very unclear. Additionally, did the numerical simulation pass the mesh test? The contour plots seem unusual. The numerical simulation should be redone very carefully.

Response to Comment 7:

Thank you to Reviewer 4 for the detailed comments. In the original manuscript, Fig. 2b simulates the fluid field and the changes in the TPCL in two different systems: one with a micropillar template and one without. The black solid lines were used as aids to clearly

illustrate the recirculation in the fluid field, while the arrows indicate the direction of fluid flow. The color gradient from blue to red represents the variation in fluid velocity, from slow to fast. These are common practices in fluid dynamics simulations (Langmuir 40, 5255-5269 (2024)). In the revised manuscript, we re-simulate the fluid field, as shown in Fig. R17, and have added the following discussion.

Compared to the case without micropillars, the TPCL on the substrate side transitions from a curved configuration to a linear configuration under the influence of capillary forces (Supplementary Fig. 2a). (Page 6, Lines 4-6)

A uniform flow region on the substrate side facilitates even mass transfer (Supplementary Fig.2a)³⁸⁻³⁹. (Page 6, Lines 16-17)

Fig. R17 | Microfluidic velocity field simulations: one with a micropillar template and the other without. The lines with arrows indicate the direction of fluid flow, while the color gradient from blue to red represents the fluid velocity, with blue indicating slower flow and red indicating faster flow.

Comment 8: Scale-up Issue: There are several recent works on the fabrication of uniform QLED patterns. In this work, the authors suggest that the cover is very important. I am wondering if this method is feasible for mass production, considering the scale-up issue.

Response to Comment 8:

We sincerely appreciate the valuable comments from Reviewer 4. Currently, our method has successfully fabricated QD films with an area of 25 cm². However, there are still some limitations in microstructures cover for mass production. For example, the dewetting time of the QD solution is relatively long, which may affect production

efficiency. Placing the assembly system in a temperature-controlled environment and raising the temperature could potentially accelerate this process (<https://doi.org/10.1016/j.esci.2023.100227>).

Comment 9: How did the authors obtain the color map in Supplementary Fig. 9? The measurement method was not fully described.

Response to Comment 9:

We appreciate Reviewer 4's comments. The color map in Supplementary Fig. 9 (Supplementary Fig. 16) shows the positions of the EL spectra of red, green, and blue QDs in the CIE 1931 color coordinates. We obtained these color coordinates by converting the EL emission spectra of the QDs into the CIE color space (*Nature* **612**, 679–684 (2022); <https://doi.org/10.1038/s41566-024-01496-x>).

Comment 10: Why are the illuminated images blurred? Are there no sharp images? This is strange because the authors pointed out the high resolution of the LED.

Response to Comment 10:

We appreciate Reviewer 4's comments. In the revised manuscript, we have replaced the illuminated images blurred, as shown in Fig. R18a (Supplementary Fig. 21). The high-resolution illuminated images are very clear, as shown in Fig. R18b.

[FIGURE REDACTED]

Fig. R18 | EL static image of full-color Cd-QLED and high-resolution red InP-QLED.
Scale bar: 1 cm (a), 20 μm (b).

Response to Reviewer 5:

Comment 1: This manuscript reports the fabrication of QLEDs with high electroluminescent performance as quantified by external quantum efficiency (EQE) and luminance. Self-assembled layers of quantum dots were used as uniform and dense films that formed in a solvent evaporation-driven process to replace the disordered, rough films previously used. The authors report reduced charge leakage and thermal losses. The use of superlattices as functional parts of QLEDs is certainly of interest to the community, and the manuscript is over all of good quality.

The assembly process that is driven by evaporation and guided by a microstructured surface is the main novelty of the manuscript. However, the process and its template are only described briefly, with few technical details. I do not believe that the contribution should be published unless detailed data on the geometries and deposition/evaporation conditions that yield the self-assembled films are added.

More specifically, I believe that the following points should be addressed:

Response to Comment 1:

We would like to thank Reviewer 5 for their time, effort, and positive evaluation of our manuscript, especially for recognizing the evaporation-driven self-assembly strategy used to fabricate uniform and dense QD films, as well as their application in QLEDs. Based on Reviewer 5's constructive comments, we will provide more technical details on the self-assembly process in the revised manuscript to ensure clarity of this important technique. We believe that these revisions have significantly improved the manuscript and support its publication in *Nature Communications*.

Comment 2: The statement on line 51: "Currently, the irregular morphology and non-uniform particle sizes of heavy-metal-free QDs present challenges for spin-coating to prepare dense and uniform films," is misleading. Uniform heavy metal-free QDs are state of the art (see, for example, *Nat. Rev. Mater.* 2023, 8, 742-758). It is unclear whether the remaining challenges are due to QD synthesis or their assembly via spin-coating. It is confusing that the authors themselves report in this contribution that InP particles (which, according to the authors, were uniform and regular in size), yielded poor QLEDs. If the assembly is the limiting step, the abovementioned statement should be replaced.

Response to Comment 2:

We agree with Reviewer 5's comments. The statement, "Currently, the irregular morphology and non-uniform particle sizes of heavy-metal-free QDs present challenges for spin-coating to prepare dense and uniform films," indeed contains

ambiguity. In recent years, significant progress has been made in the synthesis of heavy-metal-free QDs, particularly in improving their size uniformity and morphological regularity (*Nat. Rev. Mater.* **8**, 742–758 (2023); *Angew. Chem. Int. Ed.* **55**, 3714–3718 (2016)). However, we would like to further clarify that, despite the good uniformity achieved during synthesis, factors such as solution concentration, surface tension, and evaporation rate during the spin-coating process can still negatively affect the film's uniformity and density (*Adv. Mater.* **31**, 1904610 (2019); *J. Am. Chem. Soc.* **140**, 8690–8695 (2018)).

In our manuscript, the reported InP-based QDs (with a FWHM of 39 nm) exhibited relatively uniform size distribution, but still showed a significant gap compared to Cd-based QDs (with a FWHM below 25 nm). During the spin-coating process, the drying dynamics of the QD solution are difficult to control, leading to localized particle aggregation or the formation of microvoids, which negatively affect the performance of QLEDs (*Adv. Mater.* **31**, 1904610 (2019); *J. Am. Chem. Soc.* **140**, 8690–8695 (2018)). Therefore, spin-coating remains a critical bottleneck in achieving uniform and dense films during the heavy-metal-free QD assembly process.

To address these issues, we have revised the manuscript to more accurately convey this point, and the specific discussion is presented below.

Although significant progress has been made in the synthesis of heavy-metal-free QDs, achieving uniform and dense QD films through spin-coating remains a challenge¹⁸⁻²⁰.
(Page 3, Lines 16-17)

Comment 3: The evaporation-driven self-assembly is the most important process of this contribution, but few details are provided. The process is not described at all within “Methods”. The geometry of the templates is only given as “width of 20 μm , an interval of 1 μm , and a 3 μm depression”, without defining what “depressions” are, how the structures that supposedly stabilize the assembly are shaped, and how they work. It is unclear what the “stable environment is that the authors mention. It would be important to discuss whether the evaporation rate of the solvent play any role, whether it is possible to tune the thickness of the films, and how the uniformity of the films was quantified.

Response to Comment 3:

We thank the reviewer 5 for pointing out these critical issues. In the revised manuscript, we have provided detailed descriptions in the “Methods” section regarding the evaporation-driven self-assembly process for preparing uniform and dense QD films. We also recognized that the term “depressions” was not clearly defined, so we have revised it to “depth.” Additionally, we have included SEM images of the micropillar

template, clearly labeling the width, interval, and depth of the micropillars, as shown in Fig. R19. The "stable environment" mentioned in the manuscript refers to the micro-nano space in the assembly system where the airflow is isolated from external disturbances, ensuring that the solvent evaporation rate remains low and stable, which helps form denser, more uniform, and well-ordered films.

During the self-assembly process, the solvent evaporation rate indeed plays a key role. Typically, the evaporation rate can be controlled by adjusting the temperature, where higher temperatures lead to faster solvent evaporation. We measured the roughness of QD films assembled at different temperatures, as shown in Fig. R20a. The results indicate that at 25°C or below, the QD films exhibit low roughness, while the roughness increases with rising temperatures. The thickness of the QD films can be adjusted by varying the concentration of the QD solution. We tested the thickness of films at different concentrations, as shown in Fig. R20b. The results demonstrate that the film thickness increases linearly with the concentration of the solution.

In the original manuscript, we have already demonstrated the uniformity and density of the films through multiple characterization techniques, including magnified SEM images, AFM roughness maps, GISAXS patterns, and PL emission intensity maps. These results show that QD films prepared using the evaporation-driven self-assembly strategy are smooth, well-ordered, dense, and uniform, which contributes to improving the overall performance of QLED devices.

To improve the quality of the manuscript, we have made the following changes in the revised manuscript.

Based on regions with an error rate below 5% (Supplementary Fig. 2c), we designed a micropillar template with a width of 20 μm , a spacing of 1 μm , and a depth of 3 μm for QD self-assembly (Supplementary Fig. 3). (Page 6, Lines 11-14)

At temperatures of 25°C or lower, the solvent molecules are mainly diffused, and we can observe that the roughness of the prepared QD films is lower. The thickness of the film can be adjusted by varying the concentration of the QD solution (supplementary Fig. 4). (Page 6, Lines 17-20)

Fig. R19 | SEM image of a micropillars template. The width of the micropillar template is 20 μm , an interval is 1 μm , and a 3 μm depth. The edge has a reverse bending shape.

Fig. R20 | The influence of temperature (solution concentration) on the roughness (thickness) of the QD film. **a**, Relationship curve between ambient temperature and roughness of QD films. At 25 $^{\circ}\text{C}$ and below, the surface roughness of QD film is low, and the surface roughness increases significantly with the increase of temperature. **b**, The relationship between the thickness of the QD film and the concentration of QD solution. The film thickness increases almost linearly with the concentration of QD solution.

Comment 4: The authors selected red-emitting InP quantum dots but do not explain their choice. They state that “QD films prepared by the evaporation-driven strategy are denser and more uniform” (lines 168-169), but it is unclear what the reference is.

Response to Comment 4:

We sincerely appreciate Reviewer 5's detailed comments. On one hand, red InP-based QDs demonstrate superior performance compared to green InP-based and blue ZnSeTe-based QDs in the realm of heavy-metal-free QDs (*Adv. Funct. Mater.* **34**, 2313811 (2024); *Adv. Optical Mater.* **11**, 2300659 (2023); *Adv. Funct. Mater.* **32**, 2204529 (2022)). On the other hand, they are still quite far from commercialization for practical applications. Therefore, it remains crucial to further develop strategies that enhance

their efficiency, luminance, and stability.

In the original manuscript, we have provided multiple pieces of evidence supporting the higher density and improved uniformity of QD films prepared using the evaporation-driven strategy. For instance, SEM images clearly show a denser arrangement of QDs; the PL intensity distribution maps indicate enhanced uniformity of the films; furthermore, the distinct diffraction spots in the GISAXS images demonstrate the ordered arrangement of QDs in the film. The key to these results lies in our precise control of the TPCL on the substrate, maintaining its linearity and guiding its directional motion. This allows the QDs to self-assemble at the TPCL, resulting in denser, more uniform, and more ordered QD films. Numerous studies have also shown that controlling the directional movement of the TPCL is an effective method to achieve long-range ordered arrangement of nanoparticles (*Adv. Mater.* **29**, 1703143 (2017); *Adv. Mater.* **34**, 2106857 (2022); *Adv. Mater.* **36**, 2314061 (2024)).

To further enhance the manuscript, we have added a discussion on the reasons for selecting red InP-based QDs. The revised content is as follows:

We chose red InP-based QDs for film fabrication (Supplementary Fig. 5) because they represent the most promising alternative to Cd-based red QDs. (Page 6, Lines 21-22)

Comment 5: The authors report record-breaking EQE and luminance as summarized in Supplementary Table 1. A comparison directly in this table would aid the reader and underline the superior performance.

Response to Comment 5:

We fully agree with Reviewer 5's suggestion that directly comparing QLED performance in Supplementary Table 1 will enhance clarity and highlight the superior performance of our devices. Supplementary Table 1 clearly shows EQE and luminance values for similar devices, which provides a comprehensive view of the results and highlights the superior performance of our QLEDs.

Comment 6: The Methods section does not clearly state whether the protocols used in the preparation of InP QDs were novel or adapted from state of the art, and references to the base processes are missing. The same is true for the fabrication of the QLEDs.

Response to Comment 6:

We sincerely thank Reviewer 5 for raising this critical point. The red InP-based QDs used in this study are commercially available, and their source has been clearly indicated in the Methods section. Additionally, the fabrication processes of both QLEDs and patterned QLEDs have been further elaborated and expanded in the Methods section. The specific revisions are as follows:

Fabrication of QLEDs. The fabrication process of QLEDs comprises the following steps: Initially, a thorough cleaning of the indium tin oxide (ITO) glass substrate is conducted using a sequence of continuous cleaning processes involving deionized water for 30 min, acetone for 15 min, isopropanol for 15 min, and ozone treatment for 20 min. Subsequently, PEDOT:PSS (CLEVIOS P VP AI 4083; solvent: water) is uniformly spin-coated onto the ITO glass substrate at a speed of 4000 rpm, followed by a 30 min baking at 150 °C in ambient air. Next, the ITO substrate coated with PEDOT:PSS is transferred into a nitrogen-filled glovebox, where a TFB solution with a concentration of 8 mg mL⁻¹ (solvent: chlorobenzene) is spin-coated onto the PEDOT:PSS layer at 2000 rpm. This is followed by a 20 min baking step at 130 °C under a nitrogen atmosphere to form a uniform TFB layer. Within the glovebox, the evaporation-driven QD self-assembly strategy is employed to uniformly and densely deposit QDs (concentration: 8 mg mL⁻¹; solvent: octane) onto the TFB layer. In conventional QLEDs, the emissive layer is typically formed by spin-coating a QD solution with a concentration of 15 mg mL⁻¹ (solvent: octane) onto the TFB layer at 3000 rpm, followed by a 5 min baking step at 100 °C. Subsequently, Zn_{0.85}Mg_{0.15}O NPs (30 mg mL⁻¹; solvent: anhydrous ethanol) are spin-coated onto the QD layer at 3000 rpm, followed by a 30 min baking at 60 °C to create the Zn_{0.85}Mg_{0.15}O NPs layer. Finally, the multi-layer structure is placed into a dedicated high-vacuum thermal evaporation chamber, where a patterned Al cathode layer with a thickness of 100 nm is deposited via in-situ masking, resulting in a QLED with an effective emissive area of 3 mm². (Page 14, Lines 7-26)

Fabrication of patterned QLEDs. The fabrication process of patterned QLEDs is similar to that of conventional QLEDs. First, PEDOT:PSS and TFB layers are sequentially deposited onto the ITO substrate via spin-coating. Then, photoresist is spin-coated at a speed of 4000 rpm for 30 s and thermally treated at 90 °C for 90 s. The sample is subsequently transferred from the glovebox to a photolithography machine, where a patterned or pixelated mask is applied for 5 s of UV exposure. The sample is then transferred back to the glovebox, developed in hexane for 20 s, rinsed in ethyl acetate for 20 s, and thermally treated again at 150 °C for 5 min to form a patterned photoresist layer. Next, by employing the evaporation-driven QD self-assembly strategy, QDs are uniformly and densely deposited in the exposed areas of the photoresist pattern and on the TFB surface. Subsequently, a Zn_{0.85}Mg_{0.15}O nanoparticle layer and Al electrode are deposited, completing the fabrication of the patterned QLED device. (Page 14, Lines 27-36; Page 15, Lines 1-2)

Comment 7: The following two comments are suggestions and based on curiosity; the authors may choose whether to answer/consider them.

Response to Comment 7:

We greatly appreciate the curiosity and exploratory spirit you demonstrated during the review process. We will carefully consider these suggestions and assess their potential impact and possibilities for improving our research. Although these suggestions are non-mandatory, we believe they provide important insights that can help further refine our work.

Comment 8: The stability of the fabricated QLEDs was analyzed. I was wondering whether the acidic PEDOT:PSS corrodes the ITO electrode.

Thank you, Reviewer 5, for raising this thought-provoking question. The stability of QLEDs, particularly blue QLEDs, is indeed a critical challenge for their commercialization. The issue of acidic PEDOT:PSS corroding the ITO electrode has been extensively discussed in the literature (*Adv. Optical Mater.* **12**, 2301559 (2024); *Mater. Horiz.* **7**, 1759-1772 (2020)). This corrosion could potentially impact the overall stability of QLEDs. However, the stability of QLEDs is not solely dependent on the interaction between PEDOT:PSS and ITO. It also involves multiple factors, such as the structural compatibility between the QDs and the device architecture, the efficiency of electron-hole recombination, the suppression of charge leakage, heat management, and the operating environment (*Nat. Nanotechnol.* **18**, 1168–1174 (2023); *Nat. Photon.* **16**, 505–511 (2022)). While PEDOT:PSS may have some effect on the ITO electrode, it remains the most widely used hole injection layer in QLEDs. Many high-stability QLED devices still employ PEDOT:PSS as the hole injection layer (*Nat. Photon.* **18**, 186–191 (2024); *Nat. Commun.* **15**, 783 (2024); *Nat. Commun.* **15**, 5161 (2024)), so we believe it is not appropriate to attribute the stability issues solely to the corrosion of ITO by PEDOT:PSS. It is crucial to continue exploring and optimizing other contributing factors to further enhance the stability of QLEDs.

Comment 9: Blue-emitting QDs (InP would be interesting because it does not contain Cd) are of great practical interest. Did the authors test whether their concept applies to them?

Response to Comment 9:

We strongly agree with the reviewer 5's comment. Given that the performance of InP-based blue QLEDs (peak EQE of 2.6%) (*Adv. Optical Mater.* **10**, 2200685 (2022)) falls far short of that of ZnSeTe-based blue QLEDs (peak EQE of 20.2%) (*Nature* **586**, 385–389 (2020)), we selected ZnSeTe-based blue QDs as the subject of our study. ZnSeTe-based blue QLEDs are currently the most extensively researched heavy-metal-free blue QLEDs, making them highly relevant for our investigation (*Adv. Funct. Mater.* **34**, 2313811 (2024); *Chem. Eng. J.* **489**, 151347 (2024); *Adv. Mater.* **35**, 2303528 (2023);

Small, **19**, 2303247 (2023)).

SEM images (Fig. R21a) indicate that following the evaporation-driven self-assembly process, ZnSeTe-based blue QDs formed a uniform and dense film structure. Subsequently, we integrated this ZnSeTe-based blue QD film into QLED devices for electrical performance testing, where we observed trends similar to those in red InP-based QLEDs. As shown in Fig. R21b-c, in the ohmic contact region, QLEDs based on densely packed uniform QD films exhibited lower current densities; at the same current density, these QLEDs displayed higher brightness, leading to a higher EQE. However, due to the current scarcity of high-quality ZnSeTe-based blue QD materials on the market, although we observed performance improvements, the overall performance did not reach the current state-of-the-art levels (*Nature* **586**, 385–389 (2020)). Consequently, these results were not included in the initial manuscript. In response to your comments, we will incorporate these experimental results and relevant data into the revised manuscript. We have made the following revisions to the original manuscript.

To confirm the versatility of the evaporation-driven self-assembly strategy, we fabricated QLEDs by self-assembly of InP-, ZnSeTe- and Cd-based QDs. All of these devices shows significant enhancement of performances compared to the spin-coated films (Supplementary Figs. 17-18). (Page 11, Lines 10-13)

Fig. R21 | Uniform QD films prepared through the evaporation-driven self-assembly strategy resulted in enhanced electrical performance for blue ZnSeTe-based QLED. a, SEM images of blue ZnSeTe-based QD films prepared via the evaporation-driven self-assembly strategy. The images highlight the uniformity of the QD films, demonstrating the versatility of this self-assembly approach in fabricating high-quality QD films across different compositions. **b-c,** Current density-luminance-voltage (**b**) and EQE-current efficiency-current density (**c**) characteristics of blue ZnSeTe-based QLEDs. QLEDs with uniform QD films show a significant reduction in current density within the ohmic contact region, with the EQE rising from 6.7% to 8.6% and the luminance increasing from 2.0×10^4 cd m⁻² to 2.3×10^4 cd m⁻².

Comment 10: In summary, this contribution is of interest to readers of *Nature Communication*, but it lacks crucial information on a key process step and should not

be accepted in this form. I believe that it will be possible to address this issue in a major revision.

Response to Comment 10:

Thank you for your evaluation and suggestions regarding our manuscript. We understand and acknowledge your concerns about the lack of crucial information on the key process step. We have carefully reviewed the issues you raised and have provided more detailed and comprehensive explanations in the revised manuscript to ensure that readers can fully understand our research process and results. We believe that with this major revision, the manuscript will better meet the high standards of *Nature Communications* and offer greater scientific value.

Response to Reviewer 6:

Comment 1: I co-reviewed this manuscript with one of the reviewers who provided the listed reports. This is part of the *Nature Communications* initiative to facilitate training in peer review and to provide appropriate recognition for Early Career Researchers who co-review manuscripts.

Response to Comment 1:

Thank you for your thorough review and valuable feedback on this manuscript. We are pleased to learn that you co-reviewed the manuscript with another reviewer, particularly in the context of *Nature Communications* initiative to involve Early Career Researchers in the peer review process. This initiative not only helps to enhance the overall quality of research but also provides invaluable learning and development opportunities for emerging scholars. We believe that such collaborative review processes allow for a more comprehensive evaluation of the manuscript's quality and further promote the growth and exchange within the research community. We sincerely appreciate the efforts you and your co-reviewer have dedicated to this review.

Response to Reviewers and Revised Details

Response to Reviewer 1:

Comment 1: The authors have revised manuscript according to my comments.

Response to Comment 1:

We thank the reviewer 1 very much for the inspirational comments!

Response to Reviewer 2:

Comment 1: The authors have addressed most of my concerns and have made comprehensive revisions. Some of my concerns remain unaddressed.

Response to Comment 1:

We sincerely appreciate the encouraging feedback from Reviewer 2. In response to the concerns raised by Reviewer 2, we have carefully reconsidered the points and made further revisions in the manuscript to address them. We believe these revisions have significantly improved the manuscript and support its publication in *Nature Communications*.

Comment 2: The key point of the manuscript is the realization of dense QD films by evaporative-driven self-assembly strategy. This method can reduce the leakage current and potentially enhance the lifetime of QLED. However, as I mentioned in the first round of review, the leakage current is most significant in the blue QLED due to the difficulty of charge injection and poor charge balance. Therefore, the efficiency improvement in blue QLED should be the most significant, but I did not see these results in the manuscript. Besides, the authors should demonstrate the effects of leakage current reduction and lifetime enhancement in blue QLED, rather than in red devices.

Response to Comment 2:

We greatly appreciate Reviewer 2's constructive comments. We fully agree with the reviewer's point regarding the difficulty of charge injection in blue QLEDs. However, the argument that poor charge balance and leakage current is most significant in the blue QLEDs warrants further discussion. With ongoing optimization of the blue QD composition and advancements in device fabrication technologies, numerous reports in recent years have demonstrated breakthroughs in the external quantum efficiency (EQE) of blue QLEDs, with some results approaching the theoretical maximum ($EQE_{\max} = 23\%$) (*Adv. Optical Mater.* **10**, 2200319 (2022); *Nat. Photon.* **16**, 505–511 (2022); *Nat. Commun.* **14**, 284 (2023); *Nano Lett.* **24**, 5729–5736 (2024); *Nat. Commun.* **15**, 783 (2024)). Therefore, whether charge imbalance continues to be a challenge for blue QLEDs requires more precise expression. Similarly, the assertion that leakage current is the most significant in the blue QLEDs should be revisited with caution, as some studies have shown that the current density before turn-on in blue QLEDs is not necessarily higher than that in red QLEDs (*Nat. Commun.* **14**, 284 (2023); *Nat. Nanotechnol.* **18**, 1168–1174 (2023)).

To better address Reviewer 2's concerns, we have included cross-sectional TEM images of both red and blue QLEDs, as well as spin-coated red and blue QD films (Fig. R1). In Figure R1a, we clearly observe that the emissive layer in red QLEDs consists of a

single-layer QD structure, with more nanopores within the film, resulting in a larger contact area between the electron transport layer and the hole transport layer. In contrast, the emissive layer in blue QLEDs is composed of a three-layer QD structure (Fig. R1b), with fewer nanopores, leading to a smaller contact area between the electron and hole transport layers. This may explain the significant improvement in the efficiency of red QLEDs.

Although we provided preliminary lifetime test results for blue QLEDs in our first-round response, we would like to emphasize that the main focus of this manuscript is still on red InP-based QLEDs. The efficiency improvement and lifetime issues of blue QLEDs are complex and challenging topics, and we look forward to future opportunities to collaborate with Reviewer 2 and other experts in this field to jointly tackle the bottlenecks related to the operational lifetime of blue QLEDs.

Fig. R1 | Cross-sectional TEM images of red and blue QLEDs and SEM images of red and blue QD films. a, The emissive layer in red QLEDs is composed of a single-layer QD structure, with a significant number of nanopores within the film. **b,** The emissive layer in blue QLEDs consists of a three-layer QD structure, with fewer nanopores in the film.

Comment 3: Supplementary Fig. 18, the QLED with rough film exhibits higher EQE. Please double check.

Response to Comment 3:

We greatly appreciate Reviewer 2's kind reminder. We have made the necessary revisions in the Supplementary Information (Supplementary Fig. 21) section of the revised manuscript. The specific changes are as follows:

Fig. R2 | Uniform QD films prepared through the evaporation-driven self-assembly strategy resulted in enhanced electrical performance for green InP-based and blue ZnSeTe-based QLEDs. a, SEM images of green InP-based QD film prepared via the evaporation-driven self-assembly strategy. Scale bar, 100 nm. Current density-luminance-voltage and EQE-current efficiency-current density characteristics of green InP-based QLEDs. Compared to QLEDs fabricated using conventional methods, those with uniform QD films exhibit significantly lower current density in the ohmic contact region, with the EQE increasing from 7.2% to 9.8% and the luminance improving from 3.5×10^4 cd m⁻² to 5.0×10^4 cd m⁻². **b,** SEM images of blue ZnSeTe-based QD film prepared via the evaporation-driven self-assembly strategy. Scale bar, 100 nm. Current density-luminance-voltage and EQE-current efficiency-current density characteristics of green InP-based QLEDs. Compared to QLEDs fabricated using conventional methods, those with uniform QD films exhibit significantly lower current density in the ohmic contact region, with the EQE increasing from 7.2% to 9.8% and the luminance improving from 3.5×10^4 cd m⁻² to 5.0×10^4 cd m⁻².

Comment 4: I don't understand the purpose of demonstrating the patterned QLED (Fig. 4). As I pointed out in the first round of review, this method cannot be used to pattern the red, green, blue QD EML, and thus is useless for manufacturing full-colour QLED displays. To avoid misleading the readers, Fig. 4 and related discussion should be deleted.

Response to Comment 4:

We sincerely appreciate and respect the differing opinion of Reviewer 2. First, we would like to clarify that the development of film-based QLEDs and patterning-based QLEDs represent two distinct technological paths (*Nature* **575**, 634–638 (2019); *Nature* **635**, 854–859 (2024); *Nat. Photon.* **5**, 176–182 (2011)). The development of high-performance patterned QLEDs facing more complex challenges (*Nat. Photon.* **18**, 1105–1112 (2024)).

We would like to emphasize that the development of patterned QLEDs involves not only the patterning of QD emissive materials but also the patterning of charge blocking layers (*Nat. Photon.* **16**, 297–303 (2022)). The methods for patterning QD emissive materials mainly include techniques such as inkjet printing, photolithography, and electrophoretic deposition, each of which faces different challenges. Inkjet printing must address the complex fluid dynamics affecting QD deposition, as well as the impact on the deposition of hole injection layers (typically PEDOT:PSS), hole transport layers (such as TFB, PF8Cz), and electron transport layers (typically ZnMgO) (*Adv. Mater.* **34**, 2107798 (2022); *Nano Res.* **13**, 2485–2491 (2020)). Photolithography often requires the addition of crosslinking agents (*Angew. Chem. Int. Ed.* **61**, e202202633 (2022); *Adv. Mater.* **35**, 2300546 (2023)), while electrophoretic deposition necessitates modifications to the surface ligands of the QDs (*Nat. Commun.* **12**, 4603 (2021); *Light Sci. Appl.* **13**, 273 (2024)). These modification processes often introduce uncertainties that can affect the performance of patterned QLEDs. In contrast, the method we propose effectively eliminates these uncertainties, providing a more reliable technical pathway for achieving high-performance patterned QLEDs. Although the patterned QLED method demonstrated in Fig. 4 still faces certain challenges in the fabrication of high-resolution full-color QLED displays, we believe that this method holds significant research value in enhancing the resolution, luminous uniformity, and performance of patterned QLEDs, as shown in Fig. R3.

Fig. R3 | Microscopic EL images of patterned QLED pixels fabricated by different methods. a, Patterned QLEDs fabricated via spin-coating exhibit coffee-ring effects. **b,** Patterned QLED fabricated via self-assembly strategy is free of coffee-ring effects. All

scale bar, 5 μm .

Response to Reviewer 3:

Comment 1: I co-reviewed this manuscript with one of the reviewers who provided the listed reports. This is part of the *Nature Communications* initiative to facilitate training in peer review and to provide appropriate recognition for Early Career Researchers who co-review manuscripts.

Response to Comment 1:

Thank you for your thorough review and valuable feedback on this manuscript. We are pleased to learn that you co-reviewed the manuscript with another reviewer, particularly in the context of *Nature Communications* initiative to involve Early Career Researchers in the peer review process. This initiative not only helps to enhance the overall quality of research but also provides invaluable learning and development opportunities for emerging scholars. We believe that such collaborative review processes allow for a more comprehensive evaluation of the manuscript's quality and further promote the growth and exchange within the research community. We sincerely appreciate the efforts you and your co-reviewer have dedicated to this review.

Response to Reviewer 4:

Comment 1: Overall, the authors have taken the referee's critiques seriously and made commendable efforts to strengthen their manuscript. They have shifted the main emphasis toward materials and device applications, incorporated relevant literature, provided partial quantitative data, and revised figures and descriptions. These revisions clarify the core outcomes—specifically, the formation of quantum dot (QD) thin films and the enhancement of LED performance. While these changes do improve the manuscript's persuasiveness through the inclusion of performance metrics and strengthened references, there remains a significant need for a comprehensive explanation of the fluidic mechanisms involved.

Response to Comment 1:

We sincerely appreciate Reviewer 4 for recognizing our efforts and providing constructive comments. We have carefully conducted theoretical analysis on the fluid mechanics responsible for the formation of QD thin films in the revised manuscript. Our method achieves uniform and dense QD film formation mainly due to the following mechanisms.

- 1) A straight and stable retracting three-phase contact line (TPCL) at the bottom substrate created by the straight micropillars at the top template.
- 2) The deposition and assembly of QDs only at the bottom substrate facilitated by the uneven evaporation rate along the liquid meniscus.
- 3) A small enough Peclet number (Pe) allowing the QDs to assemble ordered at the TPCL.

A straight and stable retracting TPCL

The difference in contact angle at the bottom substrate (6.7°) and top template (24°) creates an asymmetric meniscus, where the TPCL at the substrate lags behind that at the template. Under the influence of the surface tension force at the TPCL, the continuous liquid film exhibits a pinning effect at the straight edge of the micropillars. Since the liquid film is confined in a narrow gap with a thickness much smaller than the width, the retraction of liquid film becomes a 2D problem, and thus the TPCL on the TFB substrate also follows a straight line parallel to the micropillar edge. While on the substrate paired with a template without micropillars as well as the pinning effect, the TPCL exhibits a curved line due to the wall effects at two sides (Supplementary Figs. 4a). To mitigate the spreading of octane along the side walls, we designed a reverse-curved structure at both ends of the micropillar template, which finally yields a neat straight assemble edge (Supplementary Fig. 4b). As the solvents evaporate and the

liquid film dewets, the TPCL jumps from edge to edge at the micropillars and retracts in a stick-slip way, as observed during droplet evaporation process at pillared surface (Fig. R4). This stick-slip behavior results in the fluctuations of the TPCL on the bottom substrate. To better evaluate the impact of these fluctuations on QD self-assembly, we introduced the concept of error rate, defined as the ratio of fluctuation time to the total dewetting time within one stick-slip cycle. Based on regions with an error rate below 5% (Supplementary Fig. 4c), we designed a micropillar template with a width of 20 μm , a spacing of 1 μm , and a depth of 3 μm for QD self-assembly (Supplementary Fig. 5), which facilitates a straight stably advancing assemble edge (Supplementary Fig. 6).

The deposition and assembly of QDs only at the bottom substrate

Equally important, our design also makes sure the deposition and assembly of QDs only at the bottom substrate, while maintaining a clean top template as the liquid film dewets. The secret lies in the stable internal directional flow resulted by the uneven evaporation rate along the asymmetric meniscus, created by the wettability difference between top and bottom plates. It is well known that the evaporation rate is largely enhanced at the wedge of geometry near the TPCL, which increases significantly with decreasing contact angle, indicating a larger evaporation rate near the TPCL at the bottom substrate. Besides, the meniscus near the top template is further from the ambient and thus need to overcome a higher vapor concentration in comparison, further lowering down its evaporation rate (Fig. R5). As the evaporation flux across the meniscus divergences at the bottom substrate, an internal capillary flow is triggered towards the bottom meniscus to compensate the liquid loss. More importantly, the uneven evaporation along the meniscus creates a temperature gradient from bottom to top and thus yields an opposite surface tension gradient, which triggers a downward interfacial flow along the meniscus, denoted as thermo-Marangoni flow (Fig. R5). Under the influences of the directional evaporation-driven capillary flow and the thermos-Marangoni flow, the particles well dispersed in the solvent enrich at the wedge region near the bottom TPCL, depositing and assembling as the TPCL retracts (Fig. R6). However, different from the chaotic internal flows with uncontrollable and non-regular TPCL (assembly edge) in droplet evaporation, our design facilitates a directional and stable 2D internal flow and assembly edge which makes sure even mass transfer along the width direction (Supplementary Fig. 4a).

A small enough Pe allowing the QDs to assemble ordered at the TPCL

The particles near the TPCL can assemble in order if they have enough time to arrange themselves within the strain by Brownian motion. The comparison between the Brownian time scale (r_p^2/D_p) and the flow time scale (L/U) can be represented by the Peclet number $Pe = Ur_p^2/(lD_p)$. Here, r_p is the particle radius, l is the initial inter-particle

spacing, $D_p = k_B T / (6\pi\mu r_p)$ is the particle diffusivity calculated by the Stokes-Einstein relation, and U is the characteristic flow velocity represented by the TPCL retraction velocity ($\leq O(10^{-7})$ m/s). $k_B = 1.38 \times 10^{-23}$ J/K is the Boltzmann constant. In our experiments, the temperature $T = 298.15$ K, the particle radius $r = 5$ nm, the viscosity of octane $\mu = 0.509$ Pa s.

$$l = 2\left(\frac{\pi}{6\phi_p}\right)^{1/3}r_p = 1.6 \text{ nm}$$

with the initial particle concentration $\phi_p = 0.001$. We thus obtain $Pe \leq O(10^{-5}) \ll 1$, which means that the QDs indeed have enough time to form an ordered film.

To strengthen our manuscript, we have supplemented these additional results in the revised manuscript. Related discussion has been included accordingly.

Fig. R4 | Schematic diagram of the dewetting process of the solution.

Fig. R5 | The evaporation flux gradually decreases along the black curve direction ($J_t > J_m > J_b$). The particles move toward the bottom substrate.

Fig. R6 | Simulation of the volume fraction of solid particles and thermo-Marangoni flow in the liquid film.

Page 6, Lines 1-22: To identify the role of micropillar templates in high-quality QD film preparation, we simulated the dewetting process of the solution (Supplementary Fig. 3). The difference in contact angle at the bottom substrate and top template creates an asymmetric meniscus, where the TPCL at the substrate lags behind that at the template. Under the influence of the surface tension force at the TPCL, the continuous liquid film exhibits a pinning effect at the straight edge of the micropillars³⁴. Since the liquid film is confined in a narrow gap with a thickness much smaller than the width, the retraction of liquid film becomes a 2D problem, and thus the TPCL on the TFB substrate also follows a straight line parallel to the micropillar edge. While on the substrate paired with a template without micropillars as well as the pinning effect, the TPCL exhibits a curved line due to the wall effects at two sides (Supplementary Fig. 4a). To mitigate the spreading of octane along the side walls, we designed a reverse-curved structure at both ends of the micropillar template, which finally yields a neat straight assemble edge (Supplementary Fig. 4b)³⁵. As the solvents evaporate and the liquid film dewets, the TPCL jumps from edge to edge at the micropillars and retracts in a stick-slip way, as observed during droplet evaporation process at pillared surface³⁶⁻³⁷. This stick-slip behavior results in the fluctuations of the TPCL on the bottom substrate. To better evaluate the impact of these fluctuations on QD self-assembly, we introduced the concept of error rate, defined as the ratio of fluctuation time to the total dewetting time within one stick-slip cycle. Based on regions with an error rate below 5% (Supplementary Fig. 4c), we designed a micropillar template with a width of 20 μm , a spacing of 1 μm , and a depth of 3 μm for QD self-assembly (Supplementary Fig. 5), which facilitates a straight stably advancing assemble edge (Supplementary Fig. 6).

Page 6, Lines 23-36; Page 7, Lines 1-8: Equally important, our design also makes sure the deposition and assembly of QDs only at the bottom substrate, while maintaining a clean top template as the liquid film dewets. The secret lies in the stable internal directional flow resulted by the uneven evaporation rate along the asymmetric meniscus, created by the wettability difference between top and bottom plates. It is well known

that the evaporation rate is largely enhanced at the wedge of geometry near the TPCL, which increases significantly with decreasing contact angle, indicating a larger evaporation rate near the TPCL at the bottom substrate³⁸⁻⁴⁰. Besides, the meniscus near the top template is further from the ambient and thus need to overcome a higher vapor concentration in comparison, further lowering down its evaporation rate. As the evaporation flux across the meniscus divergences at the bottom substrate, an internal capillary flow is triggered towards the bottom meniscus to compensate the liquid loss⁴¹. More importantly, the uneven evaporation along the meniscus creates a temperature gradient from bottom to top and thus yields an opposite surface tension gradient, which triggers a downward interfacial flow along the meniscus, denoted as thermo-Marangoni flow⁴². Under the influences of the directional evaporation-driven capillary flow and the thermos-Marangoni flow, the particles well dispersed in the solvent enrich at the wedge region near the bottom TPCL, depositing and assembling as the TPCL retracts (Supplementary Fig. 7). This phenomenon is similar to the coffee ring effect in an evaporating sessile droplet with downward interfacial Marangoni flow⁴³⁻⁴⁵. However, different from the chaotic internal flows with uncontrollable and non-regular TPCL (assembly edge) in droplet evaporation, our design facilitates a directional and stable 2D internal flow and assembly edge which makes sure even mass transfer along the width direction (Supplementary Fig. 4a) 46.

Comment 2: A higher level of numerical rigor.

Response to Comment 2:

We greatly appreciate Reviewer 4's constructive suggestions. To ensure the reliability of the numerical simulation results, we have included a grid independence test in the revised manuscript (Fig. R6) to verify the stability of the simulation results after grid refinement, thereby eliminating the impact of grid discretization on the computational outcomes. We compared eight different grid densities: ultra-coarse, coarse, standard, fine, very fine, and ultra-fine grids. For each grid type, simulations were conducted under the same conditions, and the average fluid field velocity for each grid was compared. The results show that as the grid is refined from the standard grid to finer grids, the average fluid field velocity gradually stabilizes, indicating that the refined grid is numerically convergent. Based on this, we selected grids ranging from fine to ultra-fine for further simulation analysis to ensure the accuracy and reliability of the results.

To strengthen our manuscript, we have supplemented these additional results in the revised supporting information.

Fig. R6 | Average velocity of the fluid field simulated under different mesh types.

Comment 3: Despite the progress made in addressing Reviewer 4's concerns, the authors' claims exhibit notable inconsistencies. For instance, they describe particles as being influenced by long-range force interactions in one section, while suggesting in another that these forces can be disregarded. This contradiction undermines the coherence of their argument and detracts from the overall persuasiveness of the manuscript. Moreover, the authors' treatment of evaporation, which is intrinsically linked to mass and phase transfer phenomena, suggests a lack of deep understanding in these areas.

Response to Comment 3:

We greatly appreciate Reviewer 4's constructive comments. We have rationalized the particle-particle interactions during particle assembly in the revised manuscript. In the bulk region far from the TPCL at the bottom substrate, QDs remain well-dispersed in the solvent (with a particle concentration of $\phi_p = 0.001$) and the gap between adjacent QDs is relatively large (typically $> 3 \text{ nm}$, corresponding to $\phi_p \geq 0.24$), with their stability primarily relying on solvation forces. While at the wedge region near the TPCL at the bottom substrate, the directional evaporation-driven flow enriches the QD particles as the evaporation rates maximize at this region. The gap between adjacent QDs is smaller (typically $< 1 \text{ nm}$, corresponding to $\phi_p \geq 0.39$), and the solvation forces weaken and can no longer maintain the stable dispersion of the QDs (*J. Chem. Phys.* **120**, 1921–1934 (2004), *J. Chem. Phys.* **120**, 9703–9714 (2004)), which can only occur at the wedge region very close to the bottom TPCL. At this small region with a high enough particle concentration, the van der Waals force between the QD ligands begin to dominate the self-assembly behavior (*Small* **5**, 1600–1630 (2009)). As the TPCL (the wedge-shaped liquid film) retracts, the QDs begin to precipitate and self-assemble, ultimately forming a uniform and dense QD film.

We have conducted theoretical analysis on the dynamics of solvent evaporation as well as the liquid flow induced by the uneven evaporation. As we explained in the response to Comment 1, the solvent evaporation mainly occurs near the TPCL at the bottom substrate, which generates an evaporation-driven capillary flow towards the bottom TPCL that compensates the liquid loss, and a thermos-Marangoni flow along the meniscus towards the bottom TPCL due to the induced surface tension gradient created by the temperature gradient. This flow enriches the QD particles at the wedge region near the bottom TPCL to facilitate assembly. The evaporation rate can be estimated as:

$$J_e = -\frac{1}{n_L k_B T} D_v \nabla p_v$$

which should equals to the retraction velocity of the TPCL. Here, D_v is the diffusion coefficient of the organic vapor in air, and ∇P_v is the gradient of the organic vapor concentration near the surface. Considering a zero vapor concentration at infinity, the pressure gradient can be estimated as $\nabla P_v \approx -P_s/L$, with P_s representing the ratio of the saturation vapor pressure of the solvent. n_L is the number density of the solvent molecules in the liquid state, k_B is the Boltzmann constant, and T is the temperature. For octane, $n_L = 3.71 \times 10^{27} \text{ m}^{-3}$, $D_v = 5 \times 10^{-6} \text{ m}^2/\text{s}$, and $P_s = 1723.2 \text{ Pa}$. Using $L = 5 \text{ mm}$, we calculated $J_e = 1.13 \times 10^{-7} \text{ m/s}$, which is of the same order of the experimental measured TPCL retracting velocity ($4.63 \times 10^{-7} \text{ m/s}$) in Supplementary Fig. 6.

The small evaporation rate in the narrow gap between two plates also makes sure a small enough Peclet number. Even for a TPCL velocity of $O(10^{-10}) \text{ m/s}$ at $L = 0.005 \text{ mm}$, we can also get a $Pe = O(10^{-2}) \ll 1$, which makes sure the QDs indeed have enough time to form an ordered film.

To strengthen our manuscript, we have supplemented these additional results in the revised manuscript. Related discussion has been included accordingly.

Page 7, Lines 9-20: The particles near the TPCL can assemble in order if they have enough time to arrange themselves within the strain by Brownian motion. The comparison between the Brownian time scale (r_p^2/D_p) and the flow time scale (L/U) can be represented by the Peclet number $Pe = Ur_p^2/(LD_p)^{25,42}$. Here, r_p is the particle radius, L is the initial inter-particle spacing, $D_p = KBT/(6\pi\mu r_p)$ is the particle diffusivity calculated by the Stokes-Einstein relation, and U is the characteristic flow velocity represented by the TPCL retraction velocity ($< O(10^{-7}) \text{ m/s}$). Using the parameters in our experiments, we obtain $Pe < O(10^{-6}) \ll 1$, which means that the QDs indeed have enough time to form an ordered film. At temperatures of 25°C or lower, the solvent evaporation as well as TPCL retraction are mild and the QDs assemble is highly ordered, and we can observe that the roughness of the prepared QD films is lower compared to that at elevated temperatures. The thickness of the film can be adjusted by varying the

concentration of the QD solution (supplementary Fig. 8).

Comment 4: Given this, it may be more appropriate for this work to be submitted to a materials science or applied technology journal, rather than one that requires a robust grasp of the underlying fluid-mechanical or physical principles. In summary, while the manuscript has improved in certain aspects, further refinement is necessary to address the mechanistic concerns raised and to ensure a consistent and coherent argument throughout.

Response to Comment 4:

We greatly appreciate the valuable feedback from Reviewer 4. In the revised manuscript, we have provided a comprehensive and in-depth explanation of the fluidic mechanisms involved, further reinforcing the rigor of the data and clarifying the interparticle interactions during the assembly process. Additionally, we have conducted a systematic theoretical analysis of solvent evaporation dynamics and the fluid flow induced by non-uniform evaporation. These improvements have significantly strengthened the manuscript's scientific validity and persuasiveness. Based on these comprehensive revisions and the enhanced depth of our study, we firmly believe that this work now possesses sufficient originality and scientific significance, fully meeting the publication standards of *Nature Communications*.

Response to Reviewer 5:

Comment 1: This report concerns the revised manuscript "Highly Efficient Light-Emitting Diodes via Self-Assembled InP Quantum Dots" that was submitted by Hui Li, Yuchen Wu, and co-authors for consideration at *Nature Communications*.

The revised manuscript and the detailed rebuttal letter addressed all concerns of the reviewers and resolved most. The only remaining concern is that the geometrical details of the micropillar template were only provided in the rebuttal (as Fig. R19) but not in the main manuscript or SI. We urge the authors to include this material for the readers.

The amended manuscript can be accepted for publication in our opinion.

Response to Comment 1:

We thank the reviewer 5 very much for the inspirational comments! The geometrical details of the micropillar template have been provided in SI (Supplementary Fig. 5).

Response to Reviewer 6:

Comment 1: I co-reviewed this manuscript with one of the reviewers who provided the listed reports. This is part of the *Nature Communications* initiative to facilitate training in peer review and to provide appropriate recognition for Early Career Researchers who co-review manuscripts.

Response to Comment 1:

Thank you for your thorough review and valuable feedback on this manuscript. We are pleased to learn that you co-reviewed the manuscript with another reviewer, particularly in the context of *Nature Communications* initiative to involve Early Career Researchers in the peer review process. This initiative not only helps to enhance the overall quality of research but also provides invaluable learning and development opportunities for emerging scholars. We believe that such collaborative review processes allow for a more comprehensive evaluation of the manuscript's quality and further promote the growth and exchange within the research community. We sincerely appreciate the efforts you and your co-reviewer have dedicated to this review.

Response to Reviewers and Revised Details

Response to Reviewer 2:

Comment 1: Although the authors have responded my questions, some of my concerns remain unaddressed. For Fig. 4, I don't understand the motivation for your patterning method. By patterning the electrode, which is quite simple, you can achieve the same result, so what is the advantage of your method? It would be better if you could state your motivation and explain the advantage of your method. Also, the main focus of this paper is the formation of dense QD film and its effect on device performance. The patterning results are not so related to the main topic of the paper. To avoid misleading the reader, Figure 4 and the associated discussion should be deleted.

Response to Comment 1:

We sincerely appreciate and respect Reviewer 2's valuable comments. The reviewer mentioned that similar results could be achieved by designing electrode patterns, but we believe this view may be misleading. If an anode pattern design is used, the current resolution is too low to meet the requirements of high-end display devices (such as AR/VR) (*Nature* **617**, 287–291 (2023); *Nat. Nanotechnol.* **19**, 638–645 (2024)). On the other hand, if a cathode patterning design is used, it would not allow independent pixel emission, thereby limiting the device's application (*Nature* **635**, 854–859 (2024)). Our method, however, enables the effective realization of high-resolution, high-performance patterned QLEDs, fully meeting the needs of home and near-eye displays. The specific advantages of this method have been detailed in the previous response letter.

As the reviewer mentioned, the core focus of this paper is the fabrication of dense QD films and their impact on device performance. The devices discussed include not only film-based QLEDs but also patterned QLEDs. The development of high-performance patterned devices is crucial for advancing QLED display technology toward practical applications (*Nat. Photon.* **18**, 1105–1112 (2024); *Nature* **640**, 62–68 (2025)). In conclusion, Fig. 4 not only demonstrates the application potential of self-assembly technology but also represents a key step in the transition from high-performance film-based QLEDs to high-performance patterned QLEDs. Therefore, we believe that Fig. 4 and its related discussion hold significant value and importance within the context of this article.

Response to Reviewer 3:

Comment 1: I co-reviewed this manuscript with one of the reviewers who provided the listed reports. This is part of the *Nature Communications* initiative to facilitate training in peer review and to provide appropriate recognition for Early Career Researchers who co-review manuscripts.

Response to Comment 1:

Thank you for your thorough review and valuable feedback on this manuscript. We are pleased to learn that you co-reviewed the manuscript with another reviewer, particularly in the context of *Nature Communications* initiative to involve Early Career Researchers in the peer review process. This initiative not only helps to enhance the overall quality of research but also provides invaluable learning and development opportunities for emerging scholars. We believe that such collaborative review processes allow for a more comprehensive evaluation of the manuscript's quality and further promote the growth and exchange within the research community. We sincerely appreciate the efforts you and your co-reviewer have dedicated to this review.

Response to Reviewer 4:

Comment 1: In the response regarding interparticle interactions, the authors explain that the influence of forces between quantum dots (QDs) varies with particle concentration—stating that at low concentrations, solvation forces dominate, while at high concentrations near the three-phase contact line, van der Waals interactions become significant. Although this explanation is fundamentally correct, the revised manuscript fails to explicitly acknowledge the ambiguity in the original draft. It would strengthen the response if the authors clearly admitted that the initial description was vague and then detailed how they have revised the explanation to resolve the inconsistency. This more transparent approach would help to clarify the rationale behind the concentration-dependent behavior of the interparticle interactions.

Response to Comment 1:

We greatly appreciate Reviewer 4's positive comment, especially the remark "*this explanation is fundamentally correct*". We acknowledge that the description in the original manuscript may have been somewhat ambiguous. It should be noted that we have always been committed to revising the manuscript based on the reviewers' comments to enhance its persuasiveness and clarity.

Comment 2: Regarding the issue raised by Reviewer 2 on the presentation of patterned QLEDs in Fig. 4, the reviewer noted that the demonstrated patterning method is unsuitable for fabricating full-color QLED displays, potentially misleading readers. In their response, the authors argued that the method holds intrinsic research value as a distinct technical pathway and therefore should be retained in the manuscript. However, this response does not fully address the reviewer's concern about possible confusion among readers. The authors should commit to including a clear statement—either in the main text or in the figure caption—that explicitly outlines the limitations of the patterning method, emphasizing that it is not applicable for full-color device manufacturing. Such a clarification would acknowledge the reviewer's point and mitigate potential misunderstandings about the method's applicability.

Response to Comment 2:

We greatly appreciate Reviewer 4's comments and fully understand the concerns raised by Reviewer 2 and Reviewer 4 regarding the manufacturing of full-color devices. In Supplementary Fig. 24, we demonstrated the feasibility of integrating full-color QLEDs at the centimeter scale using this method. To address the reviewers' concerns, we have removed the relevant discussion in the revised manuscript.